# Engineering the bone metastatic prostate cancer niche through a microphysiological system to report patient-specific treatment response
Cristina Sánchez-de-Diego[1,2,3,10], Ravi Chandra Yada[1,2,3,4,10], Nan Sethakorn [5,6], Peter G. Geiger[2], Adeline B. Ding[1], Erika Heninger [1], Fauzan Ahmed[2,3], María Virumbrales-Muñoz [1,7], Nikolett Lupsa[1], Emmett Bartels[2], Kacey Stewart[2,3], Suzanne M. Ponik [1,8], Marina N. Sharifi [1,9], Joshua M. Lang [1,9,11] ✉, David J. Beebe [1,2,3,11] ✉ & Sheena C. Kerr [1,11] ✉

Bone is the most common site of prostate cancer metastasis, leading to significant morbidity, treatment resistance, and mortality. A major challenge in understanding treatment response is the complex, bone metastatic niche. Here, we report the first patient-specific microphysiological system (MPS) to incorporate six primary human stromal cell types found in the metastatic bone niche (mesenchymal stem cells, adipocytes, osteoblasts, osteoclasts, fibroblasts, and macrophages), alongside an endothelial microvessel, and prostate tumor epithelial spheroids in an optimized media that supports their viability and phenotype. We tested two standard of care drugs, darolutamide and docetaxel, in addition to sacituzumab govitecan (SG), currently in clinical trials for prostate cancer, demonstrating that the MPS accurately replicates androgen response sensitivity and captures stromal microenvironment-mediated resistance. This advanced MPS provides a robust platform for investigating the biological mechanisms of treatment response and for identification and testing of therapeutics to advance patient-specific MPS towards personalized clinical-decision making.

Prostate cancer (PCa) is the most frequently diagnosed cancer among men on a global scale and the second leading cause of cancer-related deaths in men in the United States[1]. Metastatic prostate cancer represents the most lethal form of the disease, with over 85% of patients developing bone metastases. Androgens play a central role in PCa pathogenesis, making androgen signaling pathways a primary therapeutic target[2]. Over time, hormone therapy strategies have evolved with the emergence of more potent androgen receptor pathway inhibitors (ARPIs), which have shown improved outcomes for patients with castration-resistant prostate cancer (CRPC)[3]. Recent studies showed that combining ARPIs such as enzalutamide and darolutamide with androgen deprivation therapy (ADT)

significantly improves prostate cancer-specific survival compared to ADT alone. Despite these advances, resistance to ADT and ARPIs is universal, leading to the development of castration resistant prostate cancer (CRPC)[4]. Other treatments, including taxane-based chemotherapies, such as docetaxel, are utilized for patients who progress on hormonal therapies, though the median survival is generally <2 years[5]. In this context, antibody drug conjugates (ADCs) represent an emerging class of drugs with promising potential. Their increased specificity allows for a reduced minimum effective dose and an increased maximum tolerated dose, widening the therapeutic window[6,7]. Among these, sacituzumab govitecan (SG) is a first-in-class ADC consisting of a humanized monoclonal antibody directed against TROP-2,

[1]Carbone Cancer Center, University of Wisconsin, Madison, WI, USA. [2]Department of Pathology & Laboratory Medicine, University of Wisconsin, Madison, WI, USA. [3]Department of Biomedical Engineering, University of Wisconsin, Madison, WI, USA. [4]Department of Cellular and Molecular Biology, University of Wisconsin, Madison, WI, USA. [5]Department of Medicine, Loyola University Chicago, Chicago, IL, USA. [6]Department of Cancer Biology, Loyola University Chicago, Chicago, IL, USA. [7]Department of Obstetrics and Gynecology, University of Wisconsin, Madison, WI, USA. [8]Department of Cell and Regenerative Biology, University of Wisconsin, Madison, WI, USA. [9]Department of Medicine, University of Wisconsin, Madison, WI, USA. [10]These authors contributed equally: Cristina Sánchez-de-Diego, Ravi Chandra Yada. [11]These authors jointly supervised this work: Joshua M. Lang, David J. Beebe, Sheena C. Kerr. ✉e-mail: jmlang@medicine.wisc.edu; djbeebe@wisc.edu; skerr2@wisc.edu

which is expressed in PCa, and conjugated with SN-38, the active metabolite of irinotecan. SG is currently undergoing a Phase II clinical trial for metastatic PCa[8]. Treatment resistance has been linked to acquired genomic alterations such as the Androgen Receptor (AR) gene, as well as lineage state transitions. Further, the bone niche is rich with unique cell types distinct from other metastatic foci (e.g., osteoblasts, osteoclasts etc.), which can create a unique landscape that modulates tumor behavior and can influence the tumor's response to therapy[9,10]. Additionally, while stromal interactions are essential in cancer adaptation to hormonal therapies, the effects of bone stroma and androgen deprivation on cancer progression in bone are poorly understood. Patient-derived models and tissue-engineering will be critical for addressing these gaps[3,11].

Crosstalk between prostate epithelial cells and diverse cellular components of tumor microenvironments has been proposed as a driver of prostate tumorigenesis and treatment resistance in different pre-clinical mouse models of prostate cancer as well as two-dimensional transwell experiments[12–15]. However, the human multi-cellular bone metastatic niche has proven remarkably difficult to model due to a lack of translationally relevant tools and cellular components. Notably, PCa bone metastases are predominantly osteoblastic, a unique feature of PCa, characterized by excessive osteoblast activity resulting in a net gain in bone tissue[16,17]. However, these areas often exhibit parallel regions of increased osteoclast activity, giving rise to both osteopenic and osteodense areas within the same lesion[18]. The bone niche is a dynamic and heterogenous tissue that undergoes substantial compositional modifications influenced by various factors, such as age, exposure to therapeutic drugs, physiological changes, and various pathological states[19,20]. The trabecular bone structure, which constitutes ~20–40% of the volume of bone, harbors a predominant population of osteoblasts and osteocytes[21]. Although osteoclasts represent a relatively small fraction of bone cells (<1%), their numbers can be augmented in certain pathologies[22]. On the other hand, adipocytes are known to be one of the most abundant cell types, accounting for 15–70% of the cells within human bone marrow[23]. Macrophages typically constitute 15% to 20% of the cellular composition[24] while mesenchymal stem cells (MSCs) are in relatively less abundance[23,25]. It is worth noting that these proportions have been established primarily in healthy individuals, and the specific distribution of bone marrow populations in the context of PCa remains unclear. Moreover, within the metastatic niche, the tumor microenvironment (TME) assumes a crucial role in modulating the pathogenesis and progression of PCa through paracrine signaling and soluble factor signaling[26]. Comprising an intricate network of MSCs, adipocytes, osteoblasts, osteoclasts, fibroblasts, and endothelial cells (ECs), the TME actively secretes chemokines, cytokines, extracellular matrix proteins, and matrix-degrading enzymes[27,28]. Thus, the complex and bidirectional interactions between tumor cells and stromal cells in the microenvironment are pivotal drivers of metastasis, tumor outgrowth, and treatment resistance, particularly in PCa bone metastases[9,10]. Therefore, to accurately model biological responses in vitro, it is important to recapitulate these multicellular interactions.

The development of therapies for metastatic castration-resistant prostate cancer (mCRPC) urgently needs improved pre-clinical models and a comprehensive understanding of tumor biology within these understudied niches. However, conventional in vitro PCa models predominantly employ two-dimensional (2D) cultures, where cells grow as monocultures on non-physiological and rigid plastic substrates. This approach fails to accurately replicate molecular gradients, disrupts metabolism, and alters the production of extracellular matrix (ECM) proteins[29]. Furthermore, while in vivo animal models of cancer can provide insights into tumor growth and response to treatments, they can be cost-prohibitive and do not recapitulate human physiology[30]. More than 90% of cancer drugs that show potential in animal studies fail in human clinical trials due to safety or efficacy issues[28,31–33]. For example, the anatomy of the prostate gland in mice is fundamentally different from that in humans, which affects the translational relevance of findings[34]. This became clear when mouse models of bone metastatic prostate cancer showed improved survival with treatment of zoledronic acid that was not re-capitulated in multiple randomized

clinical trials[35–37]. To address these limitations, humanized mouse models have been developed as a promising alternative, offering a platform that better captures human-human cell interactions relevant to disease progression and drug response. One such approach, the humanized tissue-engineered bone construct (hTEBC), was created within NOD-SCID IL2rgnull (NSG) mice and was used to study experimental PC3-Luc bone metastases. PC3-Luc cells were found to preferentially grow in the hTEBC compared to murine bones. This model also demonstrated that the non-species-specific bisphosphonate zoledronic acid significantly decreased metastases in hTEBCs but not in murine femora, while Denosumab had no effect. These results highlight the importance of fully humanized models in preclinical research[38].

However, despite their advantages, humanized models are not without limitations. They often require extensive optimization, can be technically demanding and expensive to maintain[39]. In this context, three-dimensional (3D) cell cultures, including microphysiological systems (MPS), have emerged as a tool for investigating cancer biology, offering a unique approach that integrates both the tumor and microenvironmental components of PCa. Developing physiologically relevant in vitro models of the bone marrow (BM) niche is challenging due to the diversity of cell types, abundance of chemical mediators, and intricate structural organization. MPS offers a compact, single-platform solution that mimics the complex tumor microenvironment with high precision. Their small size and integrated design make them particularly well-suited for high-throughput screening, enabling researchers to test multiple conditions or drugs simultaneously. For example, a bioengineered microenvironment using patient-matched PCa-associated fibroblasts (CAFs) and non-malignant fibroblasts (NPFs) co-cultured with non-tumorigenic BPH-1 epithelial cells provided a 3D in vitro platform to systematically investigate the impact of tumor stroma on prostate cancer progression. In co-culture with CAFs (but not NPFs), BPH-1 cells exhibited a more invasive, elongated phenotype with increased motility and a directed migration pattern, highlighting the critical role of stromal interactions in cancer progression[40]. Another MPS approach combined murine calvarial pre-osteoblast cells (MC3T3-E1s) with prostate cancer cell lines like LNCaP and C4–2B to study cancer cell migration[41]. Another bioengineered bone-mimetic environment demonstrated stroma-mediated chemoresistance and replicated radium-223 localization, however, the environment only contained MSCs and osteoblasts[42]. Additionally, the human osteoblast-derived tissue-engineered construct (hOTEC) platform successfully recapitulated osteocytes encased in mineralized tissue and co-cultured with a prostate cancer patient-derived xenograft (PDX)[43]. Other micro-scale chips have shown that co-culturing MSCs with endothelial cells formed a vascularized, mineralized structure, allowing researchers to track prostate cancer cell motility[44]. Bock et al. developed an in vitro microtissue model by culturing primary human osteoprogenitor cells on melt-electrowritten polymer scaffolds to investigate the effects of bone stroma and androgen deprivation on cancer progression. Direct co-culture of AR-dependent and -independent prostate cancer cell lines (LNCaP, C4-2B, and PC3) induced molecular and functional changes similar to those observed in vivo. Notably, LNCaP cells exhibited a significant adaptive response under androgen deprivation, while the metastatic microenvironment displayed increased expression of AR, alkaline phosphatase, and dopa decarboxylase, reflecting a transition toward resistance[11]. While stromal interactions are essential in cancer adaptation to hormonal therapies, the effects of bone stroma and androgen deprivation on cancer progression in bone are poorly understood. However, current models do not fully represent the complexity of the BM niche, since they lack the integration of the multiple cell types and crosstalk necessary for accurate modeling[45]. Addressing these gaps is crucial for developing more comprehensive and physiologically relevant models that bridge the divide between preclinical studies and clinical outcomes. Precision medicine offers a promising avenue for improving treatment strategies by tailoring therapies to individual patients based on genomic profiling and targeted interventions. Recent advances in PCa precision medicine include targeting gene fusions[46], utilizing genome-editing tools[47], identifying non-coding RNA biomarkers[48],

and exploring the potential of liquid tumor profiling[49,50] in addition to the use of MPS[51]. As these technologies advance, they have the potential to transform clinical practices and improve patient outcomes[52].

In this study, we developed and optimized a MPS using the LumeNEXT[53] microfluidic platform to more accurately recreate the PCa metastatic bone TME. The platform integrates a collagen-embedded matrix of bone stromal and immune cells with spheroids derived from primary prostate epithelial cells isolated from tumor tissue. A luminal structure, lined with iPSC-derived endothelial cells, mimics vasculature and regulates endothelial behavior and growth factor profiles[54]. To validate the model, we tested the sensitivity of the system to androgen-specific responses using two standard of care drugs, darolutamide and docetaxel, before assessing the efficacy of SG in three distinct patient-derived models. We discovered insights into the metastatic bone TME and its influence on PCa treatment. Specifically, we identified that the bone TME attenuates the cytotoxic efficacy of both standard of care treatments and SG in TROP-2⁺ PCa epithelial spheroids. These findings underscore the importance of the TME in modulating treatment outcomes and reveal TROP-2-dependent responses to therapy within the metastatic bone niche. This platform provides a robust tool for studying metastatic bone disease and evaluating therapeutic strategies in physiologically relevant settings.

## Results

### Isolation and differentiation of bone stroma cell populations

The bone stroma is a complex and dynamic milieu composed of various cell types that collectively regulate the overall structure and integrity of the bone. The constant feedback and signaling between cell types play a crucial role in their development, differentiation, and phenotype determination. In our MPS, we aimed to recapitulate the major stromal cell types found in the metastatic niche of PCa, including osteoblasts, osteoclasts, adipocytes, fibroblasts, and M2 polarized macrophages. To achieve this, bone marrow mesenchymal stem cells (MSCs) were isolated from bone marrow aspirates obtained from patients with prostate cancer. MSCs were then expanded using in vitro culture and differentiated into osteoblasts, adipocytes, and fibroblasts using commercially available media formulations. Histological and immunofluorescent staining were performed to confirm the successful multilineage differential potential of MSCs (Supplementary Fig. 1). Specifically, osteogenic differentiation was assessed using Alizarin Red and Alkaline Phosphatase staining. Alizarin Red showed increased mineral matrix deposition in cells differentiated into osteoblasts compared to undifferentiated controls (Supplementary Fig. 1A). Similarly, alkaline phosphatase activity, a marker for osteoblast function, was also increased after osteoblast differentiation, further confirming successful osteoblast differentiation (Supplementary Fig. 1A). Adipogenic differentiation was determined using LipidSpot, a fluorescent probe that labels lipid droplets. After differentiation, the percentage of adipocytes, defined as cells containing lipid droplets, in each sample was above 90% while controls showed no lipid droplets (Supplementary Fig. 1B). Fibroblasts were identified using Phalloidin and fibroblast activation protein alpha (FAP) staining. Both control BM-MSCs and fibroblasts expressed FAP and Phalloidin, consistent with previous reports[55] (Supplementary Fig. 1C). Osteoclasts were differentiated from peripheral blood monocytes isolated from patients with prostate cancer, using M-CSF and RANKL. Osteoclast differentiation was assessed using tartrate-resistant acid phosphatase (TRAP) and CD163 staining. Control macrophages, differentiated using M-CSF, stained positive for both TRAP and CD163 but remained mononucleated cells. In contrast, macrophages differentiated to osteoclasts with M-CSF and RANKL fused their membranes to create polynucleated cells consistent with osteoclast differentiation (Supplementary Fig. 1D).

### High-throughput screening to develop multi-phenotype media formulation for MPS

The bone metastasis MPS is comprised of eight unique primary cell types in co-culture to recapitulate the multicellular crosstalk present in the metastatic niche. However, each of these specialized cells has specific media requirements, therefore, we needed to identify a media formulation that could support the viability and maintain phenotypes specific for each of these cell types. Furthermore, metabolites secreted by various cell types may impact the viability of other cell types, underlining the importance of multicellular co-culture media optimization. We addressed this using microDUO 384-well plate (Onexio Biosystems) screening, a high-throughput approach to identify the optimal media formulations for MPS. Unlike standard well plates, the microDUO platform incorporates a media bridge between adjacent wells, facilitating the diffusion of secreted metabolites or signaling molecules from one cell type to another without direct co-culture (Fig. 1A). This approach enables balancing the diverse requirements of each cell type in our co-culture model, while ensuring the fidelity and robustness of our experimental system.

Prostate epithelial cells, isolated from patient tumor tissue, and iPSC endothelial cells (EC), which will form the tumor and vasculature compartments of the MPS, respectively, require the highest number of tissue-specific media supplements to support in vitro growth. Therefore, we established the viability of epithelial and iPSC endothelial co-cultures in various media combinations, including different ratios of primary prostate cell epithelial culture media and iPSC endothelial cell culture media, as well as different combinations of base media formulations supplemented with additional factors listed in Supplementary Table 1. We assessed the number of viable prostate epithelial cells and iPSC ECs in co-culture using a CellTiter-Glo (Promega) assay. Viability of epithelial cells and iPSC ECs was compared to their respective monoculture conditions using their standard media formulation (Fig. 1B). Interestingly, co-culture consistently enhanced the viability of the epithelial cells across all media formulations. However, iPSC EC viability was limited in media formulations where most of the media was comprised of factors from epithelial media (media 1), while an increase was observed in media formulations containing at least 50% of epithelial base media supplemented with full endothelial cell factors (media 4 and 6). Based on these findings, we selected media 6 for further screening studies, as it supported the highest viability of both epithelial and endothelial cell populations.

Next, we evaluated the viability of M2-differentiated macrophages in co-culture with epithelial cells and iPSC ECs (Fig. 1C). To this end, we tested three additional media formulations based on media 6 and macrophage differentiation media (media 7), media 6 with additional supplementation of macrophage differentiation factors (media 8) or a mixture of 50% media 6 with 50% macrophage differentiation media (media 9) (Supplementary Table 1). Notably, none of the media combinations adversely affected the viability of the macrophages; however, media 7 significantly decreased the viability of the epithelial cells and iPSC ECs in co-culture. Consequently, we selected media formulation 8 for further experiments as it maximized macrophage cell viability while maintaining the viability of the epithelial cells and iPSC ECs (Fig. 1C). Furthermore, media 8 contained several factors used in the culture media for bone stromal cells therefore, we assessed the impact of media 8 on the viability of MSCs, fibroblasts, osteoblasts, and adipocytes and found that it supported viability across all cell types (Fig. 1D). Due to its ability to support the viability of eight distinct cell types found in our metastatic bone niche MPS, media 8 was chosen for subsequent studies in the MPS models and was renamed to "Multicellular Co-culture Media" (MCM).

We further assessed how the multicellular co-culture media influenced the phenotype of various bone cell populations after a 7-day culture period (Fig. 1E). iPSC-derived endothelial cells maintained stable CD31 expression, confirming their endothelial identity. CD163 expression, a M2 macrophage marker, remained stable, and the number of RANK⁺ osteoclasts did not change, indicating preserved osteoclast phenotype. Similarly, there was no significant difference in TROP-2, an epithelial marker, between the two media conditions. Notably, MCM maintained the TROP-2 expression heterogeneity observed in patient-derived primary cells. Osteoblast alkaline phosphatase staining, and adipocyte lipid spot staining showed no significant differences. Overall, these findings suggest that MCM effectively supported the viability and phenotype of various bone cell types.

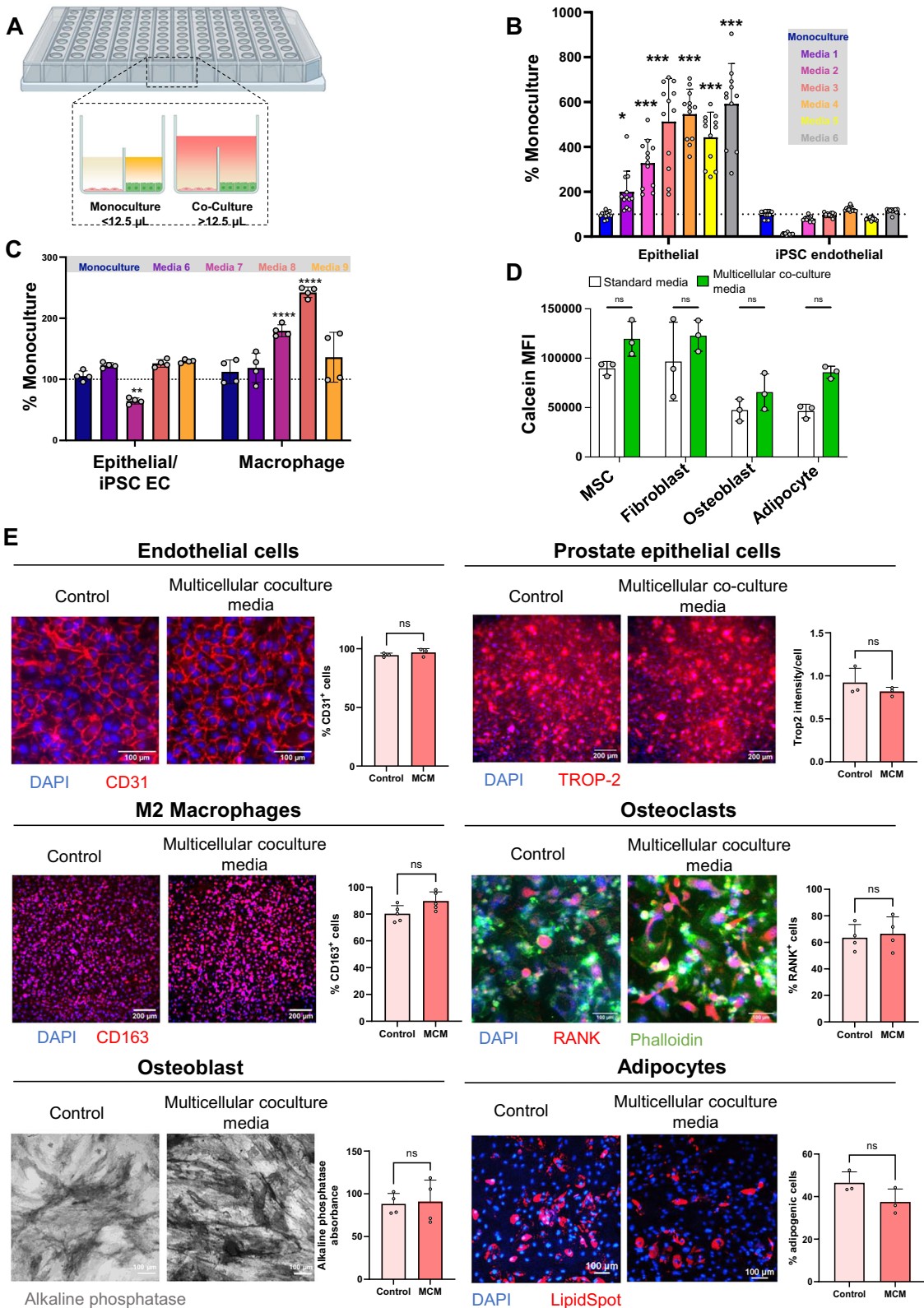

## Developing a bone microenvironment MPS

To develop a MPS of the metastatic PCa bone microenvironment that can recapitulate the multicellular crosstalk within this niche, we used our previously developed LumeNEXT platform to establish a bone microenvironment with endothelial vasculature that can be surrounded by a mixed population of bone stromal and immune cells[53,56–58]. The design and

fabrication of the LumeNEXT devices were performed as previously described, wherein two overlapping layers of PDMS form a microfluidic chamber with a removable PDMS rod sandwiched in between, serving as a mold for creating a luminal structure[53]. The central chamber of the device was filled with a matrix of collagen, the primary protein of bone ECM, along with fibronectin, and fibrinogen with embedded bone stromal cells,

**Fig. 1 | Stromal cells retain their viability and phenotype in multicellular co-culture media. A** Schematic representation of the MicroDUO platform used for media optimization, wherein, multiple cell types can be co-cultured by bridging media between wells. **B** Cell viability of epithelial cells co-cultured with iPSC endothelial cells under 6 different media formulations with respect to the mono-culture media formulation ($n = 3$). **C** Cell viability of epithelial cells and iPSC endothelial cells co-cultured with macrophages under 5 different media formulations with respect to the monoculture media formulation. ($n = 3$) Media formulations are listed in Supplementary Table 1. **D** Cell viability of BMMSCs, fibroblasts, osteoblasts, and adipocytes in monoculture under their standard culture media or the multicellular co-culture media. $n = 3$ experimental replicates. **E** Representative

images and quantification of various phenotype markers were obtained in different cell populations. Cells were differentiated and then cultured for 7 days in either standard differentiation media or multicellular co-culture media (MCM). The staining included DAPI for nuclear staining, CD31 for endothelial cells ($n = 3$); CD163 for M2 macrophages ($n = 4$); TROP-2 for primary epithelial prostate cells ($n = 4$); RANK and phalloidin for osteoclasts ($n = 4$), Alkaline phosphatase for osteoblasts ($n = 3$), and LipidSpot for adipocytes ($n = 3$). Each point indicates a technical replicate. *$p \leq 0.05$ **$p \leq 0.01$, ***$p \leq 0.001$, ****$p \leq 0.0001$(Student's $T$-test). Figures show mean ± SD. Scale bar represents 100 μm for endothelial, osteoclast, osteoblast, and adipocyte cell types, and 200 μm for prostate epithelial cells and M2 macrophages. Figure created with BioRender.com.

including adipocytes, osteoblasts, osteoclasts, fibroblasts, M2 macrophages, and mesenchymal stem cells, all of which were obtained from the differentiation of MSCs or monocytes, as described previously (Fig. 2A, B). Despite previously reported PDMS adsorption issues with PDMS microfluidic devices, we have not observed any obvious changes in protein partitioning in our 3D hydrogel environment within the MPS[59]. The proportions of each cell type in the MPS were determined based on their previously reported relative abundance in vivo (Fig. 2A)[20–23,25]. Following the polymerization of the hydrogel, the rod was subsequently removed, and iPSC derived endothelial cells were seeded into the lumen of the device to form an endothelial microvessel (Fig. 2C). Confocal images confirmed the presence of stromal cells within the main chamber and the formation of an endothelial microvessel expressing the CD31 endothelial marker within the MPS (Fig. 2D–F).

To recreate the dense bone marrow microenvironment while maintaining cell viability, we seeded 10,000 bone stromal cells, as detailed in Fig. 2A, into each device. To assess the platform's capacity for co-culture, viability was assessed at day 4 and 7, which consistently showed high viability across all conditions (Fig. 3A). Although cell proliferation analysis revealed no significant increase in cell numbers between days 4 and 7, the cells appeared more elongated by day 7 (Fig. 3A, B), suggesting enhanced adaptation to the microenvironment and improved cell-cell and cell-matrix interactions.

We used single-cell RNA sequencing (scRNA-seq) to characterize the cell populations in the device present after 4 and 7 days of cell culture (Fig. 3B, C). MPS were digested using collagenase to release the cells and fixed in paraformaldehyde prior to processing with the Flex protocol (10X Genomics) and next-generation sequencing analysis. To analyze the scRNA-seq data, a clustering resolution parameter of 0.35 was chosen to capture the transitional nature of many cells, as they expressed markers typical of both progenitor and differentiated states. Harmony integration was used to separate housekeeping and cell-cycle-associated genes, resulting in distinct clusters.

Cluster annotation was conducted manually using canonical markers from literature and PanglaoDB because of lack of reference datasets from bone samples in automated methods such as SingleR or scType. The cell type identification utilized specific markers to distinguish between cell types in the dataset. Endothelial cells were identified by $VWF$[60], while $COL1A1$ served as the marker for osteoblasts[61], and $ADIPOQ$ indicated adipocytes[62]. The presence of MSCs was confirmed by observing a high expression of $FN1$[63] and $BGN$[64] in conjunction with $COL1A2$[65]—which highlights that these MSCs assume the trajectory consistent with osteogenic mesenchymal lineages. Macrophages were also confirmed to be present in the sample through observing high expression values of $CD9$[66] and $SRGN$[67]. Fibroblasts were recognized by the expression of $COL4A1, NDUFA4L$. These markers provided a framework for defining each cell type within the clusters, aiding in the differentiation and classification of cells in various stages of their lineage.

We observed that the endothelial cells formed a prominent cluster on the right, which remained stable between the two time points. MSCs, fibroblasts, and osteoblasts were centrally located, with MSCs appearing to serve as a central node around which other bone cell types are distributed. The fibroblast and osteoblast populations are also distinct, suggesting

maintained identities within the co-culture environment. M2 macrophages were also clearly separated, and their clustering suggests that these cells retain specific identities associated with immune function.

Notably, most of the clusters show little movement or overlap between days 4 and 7, indicating that the phenotypes of these cells are stably maintained over time within the MPS. However, we observed an increase in the number of osteoblasts at day 7, consistent with an increase of osteoblast gene markers quantified by RT-qPCR (Supplementary Fig. 2A). Additionally, some clusters displayed markers indicative of cells in transition, reflecting ongoing differentiation from MSC into other cell types such as adipocytes or osteoblasts. This included MSCs expressing $FABP4$, which were identified as transitioning toward adipocytes. This transitional state added complexity to cell state identification, as many cells expressed markers characteristic of both progenitor and fully differentiated states. Finally, iPSC-derived endothelial cells were identified as endothelial cells due to their shared expression of endothelial-associated genes such as $VWF$. Certain clusters labeled as "mixed population" expressed primarily housekeeping genes, while proliferative clusters showed gene signatures linked to various cell cycle stages. Adipocytes were not detected in the sequencing dataset, likely due to the challenges associated with capturing this population through single-cell sequencing[68,69]. However, within the "mixed population" clusters, a subset of cells showed expression of adipocyte-associated genes, such as $LPL$ and $LEP$, although these genes were only present in approximately half of the cells within these clusters. To further confirm the presence of adipocyte-like cells in the MPS, we used LipidSpot staining to identify lipid droplets, a characteristic feature of early adipocyte differentiation. This staining revealed that roughly 30% of the total cell population in the MPS was positive for lipid droplets, indicating an early adipocyte differentiation stage (Fig. 3D). Furthermore, we detected adipocyte markers by RT-qPCR, but did not observe significant changes over the 7-day period, suggesting a stable presence of adipocyte-like cells within the culture (Supplementary Fig. 2A).

The "mixed population" cluster was further analyzed as a subcluster with a resolution of 0.25 (Supplementary Fig. 3A). The analysis involved identifying highly variable genes and applying standard preprocessing steps to determine potential marker genes for this cluster. While principal component analysis showed high variability in adipocyte-associated genes such as $LPL, LEP,$ and $FABP4$, their expression was negligible. However, a subcluster exhibited $ADIPOQ$ expression at day 7. Additionally, gene expression analysis revealed the downregulation of $COL5A1$ and $CTHC1$, which are associated with mesenchymal stem cells that typically differentiate into fibroblasts. Some cells expressed $BGN$ and $COL1A1$, markers of early osteoblast differentiation as well as fibroblasts and extracellular matrix (ECM) remodeling (Supplementary Fig. 3B). We also observed varying levels of $SRGN$ expression, a marker for hematopoietic and immune cells, while some cells expressed $VWF$, an endothelial cell marker (Supplementary Fig. 3B). The heterogeneous gene expression within this cluster makes it difficult to classify a distinct cell type. However, the downregulation of genes associated with mesenchymal stem cells, endothelial cells, and adipocytes suggests that this cluster represents a mixed population of cells that did not follow a specific differentiation trajectory or may have been damaged. This is further supported by the RNA counts and gene expression levels, which are lower than in other clusters but still above standard QC thresholds.

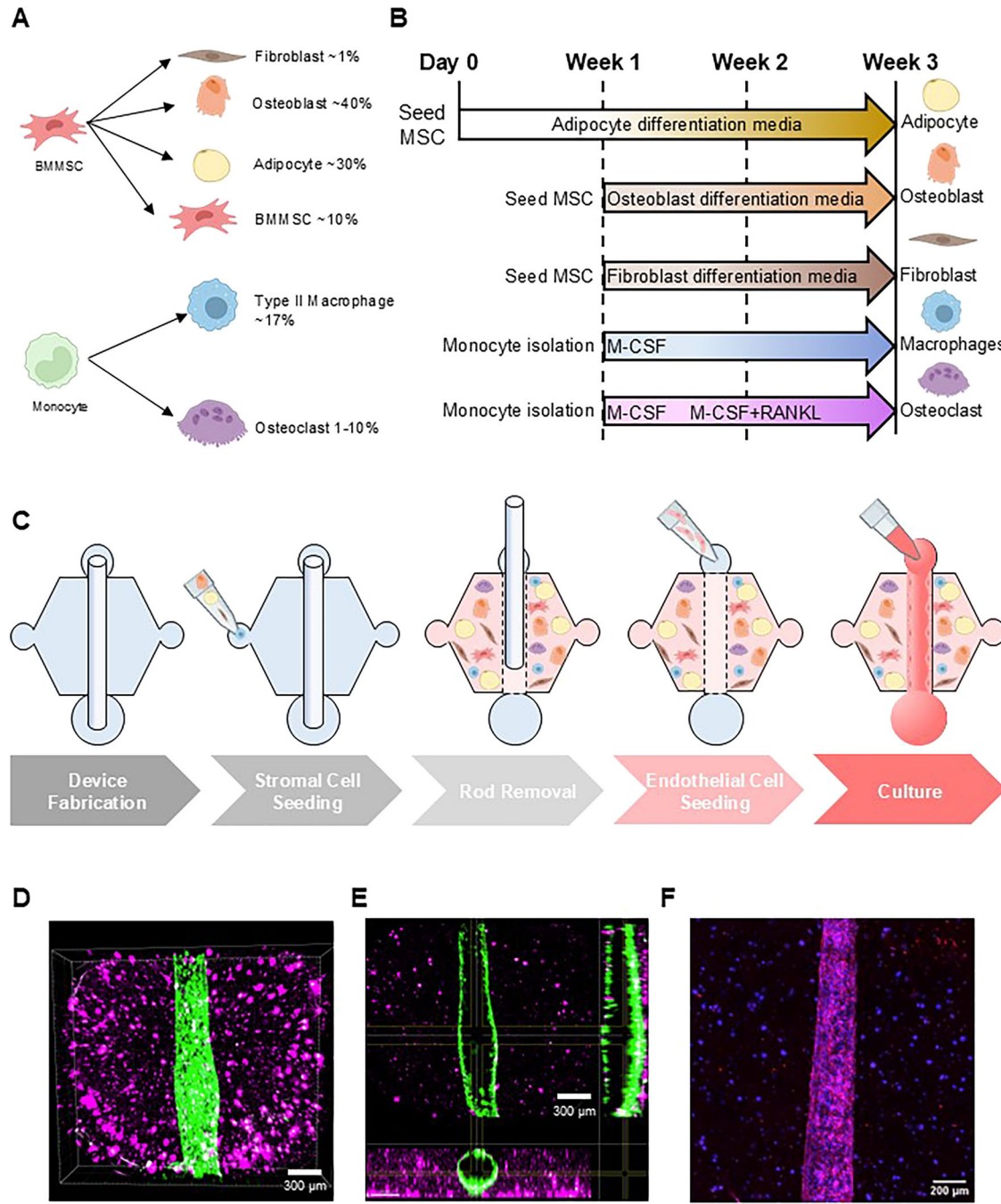

**Fig. 2 | Device illustration and representation of the cell differentiation process.** **A** Visual representation of the cell types used in this model and their relative abundance in the tumor microenvironment along with a **B** schematic representation of the differentiation process for cell types including in the bone metastasis MPS. **C** Schematic representation of device assembly. The LumeNEXT devices are fabricated with two layers with a central chamber housing a PDMS rod. The main chamber is filled with a stromal cell/ECM hydrogel solution and allowed to polymerize at RT for 30 min. Removal of the rod enables seeding of iPSC-endothelial cells in the lumen. Finally, media is added to enable co-culture of all seven cell types. **D** 3D image and **E** 2D projection of the MPS where stroma cells are stained in magenta, and lumen is stained in green. Scale bars: 300 μm. **F** Immunofluorescence staining of CD31 (magenta) and DAPI (Blue). Scale bars: 200 μm. Figure created with BioRender.com.

Osteoclasts were also not detected in the sequencing dataset, likely due to their low initial abundance within the MSC-derived populations, which may have limited their capture. Additionally, some canonical osteoclast markers may have been filtered out during preprocessing, despite efforts to retain as many cells as possible. Therefore, to confirm the presence of osteoclasts, we stained the devices for RANK at day 4 and day 7 (Fig. 3E). We found that RANK+ cells accounted for 2%–9% of the total cell population in

the MPS. These cells exhibited multinucleation, a characteristic feature of osteoclasts. Although this percentage is higher than our initial estimation, the low number of cells used in the system makes precise control challenging. Furthermore, since the starting population includes a mix of osteoclast precursors and M2 macrophages, variability in differentiation rates across patient-derived samples likely contributes to differences in the final osteoclast numbers. Additionally, we assessed osteoblast activity by measuring

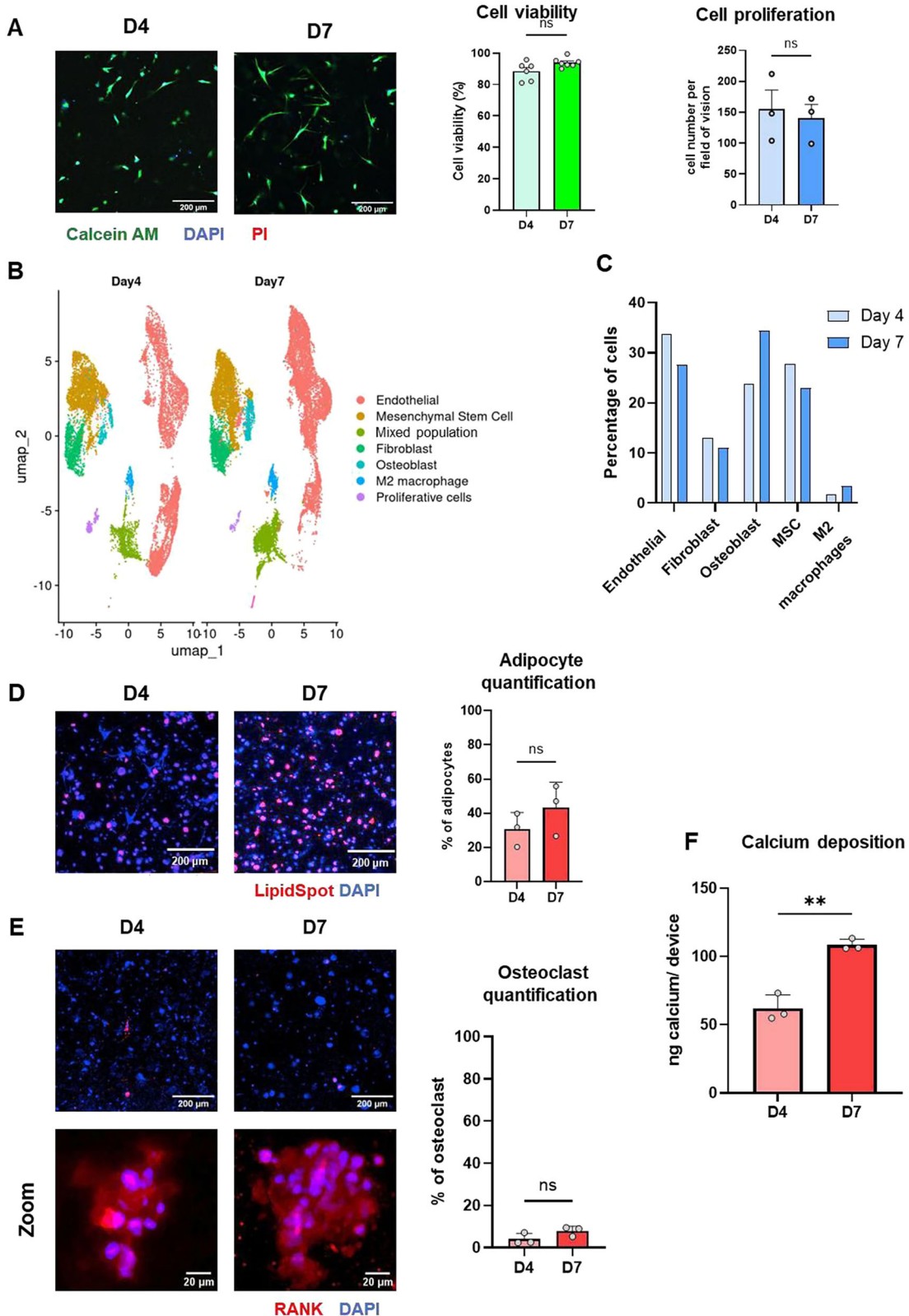

mineralization using the Stanbio Calcium LiquiColor kit, a colorimetric plate-based assay that quantifies calcium deposition as an indicator of bone matrix formation (Fig. 3F). A progressive increase in mineralization over the 7-day period suggests that osteoblasts are actively depositing calcium, demonstrating their involvement in bone matrix formation. We analyzed collagen fiber size and alignment on second harmonic generation images of

the MPS, using the CT-fire and Curve Align software packages. We observed significant remodeling of the collagen fibers in the extracellular matrix. By day 7, the collagen fibers were significantly smaller and thinner, which may suggest increased collagen degradation or restructuring, processes typically associated with active tissue remodeling (Supplementary Fig. 2B). This collagen remodeling could be indicative of dynamic

**Fig. 3 | Stromal cells retain their viability across seven days in the MPS devices.** **A** Representative images and quantification of stromal cell viability (Calcein A.M; green, PI: red, DAPI:blue) and cell number per field of vision at 10X magnification (Calcein AM; green) across 7 days. *n* = three experimental replicates *\*p* ≤ 0.05, *\*\*p* ≤ 0.01 (T-Test + Mann-Whitney test). Scale bars: 200 μm. **B** UMAP plots showing Harmony integration of single-cell RNA sequencing data at days 4 and 7. Clusters correspond to specific cell types identified by marker genes: endothelial cells (*VWF*), MSC (*POU5F1*), unknown, fibroblasts (*COL4A1*), osteoblasts (*COL1A1*), M2 macrophages (*CD9 and SRGN*), and proliferative cells. C. Relative percentage of the different cell types discovered in the MPS at day 4 and 7 by single cell sequencing.

**D** Representative images and quantification of LipidSpot staining (red) for lipid droplets in early adipocyte differentiation, with DAPI (blue) marking nuclei. Scale bars: 200 μm. *n* = 3 experimental replicates. **E** Representative images and quantification showing RANK staining (red) for osteoclasts, with DAPI (blue) marking nuclei. Scale bars: 200 μm. *n* = 3 experimental replicates. **F** Calcium deposition at day 4 and 7 determined using the colorimetric assay, Stanbio Calcium (CPC) Liqui-Color® test. *n* = 3 experimental replicates. ns = *p* > 0.05, *\*\*p* ≤ 0.01 (T-Test + Mann–Withney test). Each point indicates a technical replicate. Figures show mean ± SD.

interactions between osteoblasts, osteoclasts, and other cells in the system, reflecting a more physiologically relevant bone microenvironment.

To validate the single-cell data results, we performed qPCR to confirm molecular characterization of the cell populations within the device. We isolated RNA from six pooled devices at days 0 and 7. Through qPCR analysis we examined the gene expression profiles of all the included cell populations. Notably, we observed a significant increase in the expression of genes associated with osteoblasts (*ALPL, SP7, COL1A1*), fibroblasts (*VIM, FAP*) and osteoclasts (*ACP5, CTSK*) over time, while gene markers for MSC (*POU5F1, NANOG*) remained unchanged (Supplementary Fig. 2A). In addition, we detected osteoclast gene expression by qPCR, which increased over the 7-day period (Supplementary Fig. 2A) validating the presence of osteoclasts within the device despite their low abundance in the single-cell sequencing data. These findings validate the phenotypic assessment of osteoclast and adipocyte proportion and suggest that all the stromal cell types in co-culture remain functionally active and continue their differentiation process during in vitro cell culture. Furthermore, these results support that the bone metastasis MPS sustains the viability and phenotypic characteristics of the diverse cell populations within the stromal compartment.

### Creating the bone metastasis MPS

Following the characterization of stromal cells within the bone microenvironment MPS, our next step was to replicate the complexity of the bone metastatic niche by incorporating prostate tumor epithelial spheroids within the hydrogel. Primary prostate epithelial cells were isolated from tumor tissue specimens and expanded in culture. To characterize these epithelial cells, we performed flow cytometry to analyze the expression levels of epithelial markers (EpCAM, CD49f) and prostate-specific biomarkers (PSMA, PSA, AR) in three different primary donor samples, as well as several PCa cell lines with known expression patterns (LNCaP, LAPC4, DU-145), for comparison (Fig. 4A and Supplementary Fig. 4). Notably, all isolated primary prostate cells expressed markers characteristic of epithelial cells (e.g., EPCAM, and CD49f). DU-145 is known to be AR-negative, therefore, they served as negative controls for PSMA, PSA, and AR analyses. Expression of PSA, PSMA, and AR in the primary prostate cells was comparable to that detected in the AR-positive LNCaP and LAPC4 cells. TROP-2 expression was also measured since later studies would target the antigen. The cell lines had varying levels of TROP-2 expression, ranging from low (LNCaP), medium (DU-145), and high (LAPC4). All primary cells tested showed expression of TROP-2, which varied from medium (Donor 1 and 2) to high (Donor 3). These analyses show that the primary patient-derived prostate cells are epithelial cells expressing TROP-2 and AR PCa biomarkers. (Fig. 4A and Supplementary Fig. 4).

Subsequently, we tested the capacity of primary prostate cells to form spheroids using the hanging-drop method and observed successful spheroid formation after 48 h in culture. These spheroids were then incorporated into the hydrogel of the devices, alongside the stromal cells, and we assessed their viability at different time points. Encouragingly, primary prostate tumor epithelial spheroids remained viable for up to 7 days in the bone microenvironment (Fig. 4B–D), highlighting our capacity to isolate and culture these cells in a physiologically relevant model of the metastatic bone niche.

### Prostate tumor spheroids modify gene expression and protein secretion in the bone metastasis MPS

To investigate the crosstalk between prostate tumor cells and the bone stromal microenvironment, we performed a comprehensive analysis of gene and protein expression changes in the bone environment upon addition of primary prostate tumor epithelial spheroids into the MPS. We placed the spheroids in the side port to allow for separation from the stromal compartment and facilitated the analysis of each component (i.e., tumor and stromal cells) separately after co-culture. After 4 days in co-culture with and without primary prostate tumor epithelial spheroids, we harvested stromal cells from the devices and quantified gene expression using a RT2 qPCR Osteogenesis profiler array (Qiagen) to analyze genes associated with the bone microenvironment. Our analysis of 84 genes revealed significant changes in gene expression across all three donors where the presence of primary prostate tumor epithelial spheroids induced upregulation of 4 genes and downregulation of 3 genes (Fig. 5A). All gene expression changes are shown in Supplementary Fig. 5. Specifically, upon exposure to the primary prostate tumor epithelial spheroids, we observed increased expression of *ALPL*, a gene marker for osteoblast differentiation. We also noted elevated levels of *GLI1* and *TWIST1*, transcription factors associated with differentiation, proliferation, and survival. Additionally, we observed upregulation of *CTSK*, which is mainly expressed by osteoclasts that participate in bone matrix resorption and contribute to tumor invasiveness[70]. Interestingly, we observed downregulation of the TGF-β and bone morphogenic protein (BMP) family (*BMP4, ACVR1, and TGFB1*). BMPs are a diverse class of growth factor proteins that belong to the transforming growth factor-β (*TGFB1, TGFB2, TGFBR1, TGFBR2*) superfamily, and aberrant expression of various BMPs has been reported in several tumor tissues, including prostate cancer. Overall, these results highlight the complex biological signaling occurring between the primary prostate tumor spheroids and the TME.

To examine the impact of primary prostate tumor epithelial spheroids on the bone microenvironment at the protein expression level, we collected media from MPS with and without spheroids and conducted a multiplexed bead-based ELISA, assessing 15 different analytes (Fig. 5B). Media was sampled on the 3rd and 7th days of the co-culture to provide sufficient time for reciprocal signaling and protein accumulation to occur. Initial observations indicated that protein expression levels exhibited minimal changes by day 3, but more significant variations were evident by day 7. As shown in the qPCR analysis, co-culture of the prostate tumor spheroid with the bone stromal cells led to downregulation of growth factors and BMP pathways in the stroma. However, the bead-based ELISA reports data from the entire system and accounting for the primary prostate tumor epithelial spheroids contribution, protein levels of EGF and BMPs were comparable between controls and primary prostate tumor epithelial spheroids (Fig. 5B). Additionally, we observed an increase in TGF-α and BMP-9 protein levels in the devices with spheroids (Fig. 5B). Notably, the presence of the primary prostate tumor epithelial spheroids decreased the levels of M-CSF while increasing the expression of GM-CSF (Fig. 5B) at 7 days of culture. At that point, we also observed a trend towards increased TNF-α. These results suggest that the tumor modulates the inflammatory signaling in the bone microenvironment. The devices with spheroids exhibited lower levels of the matrix metalloprotease MMP-2, in line with the qPCR findings (Fig. 5B).

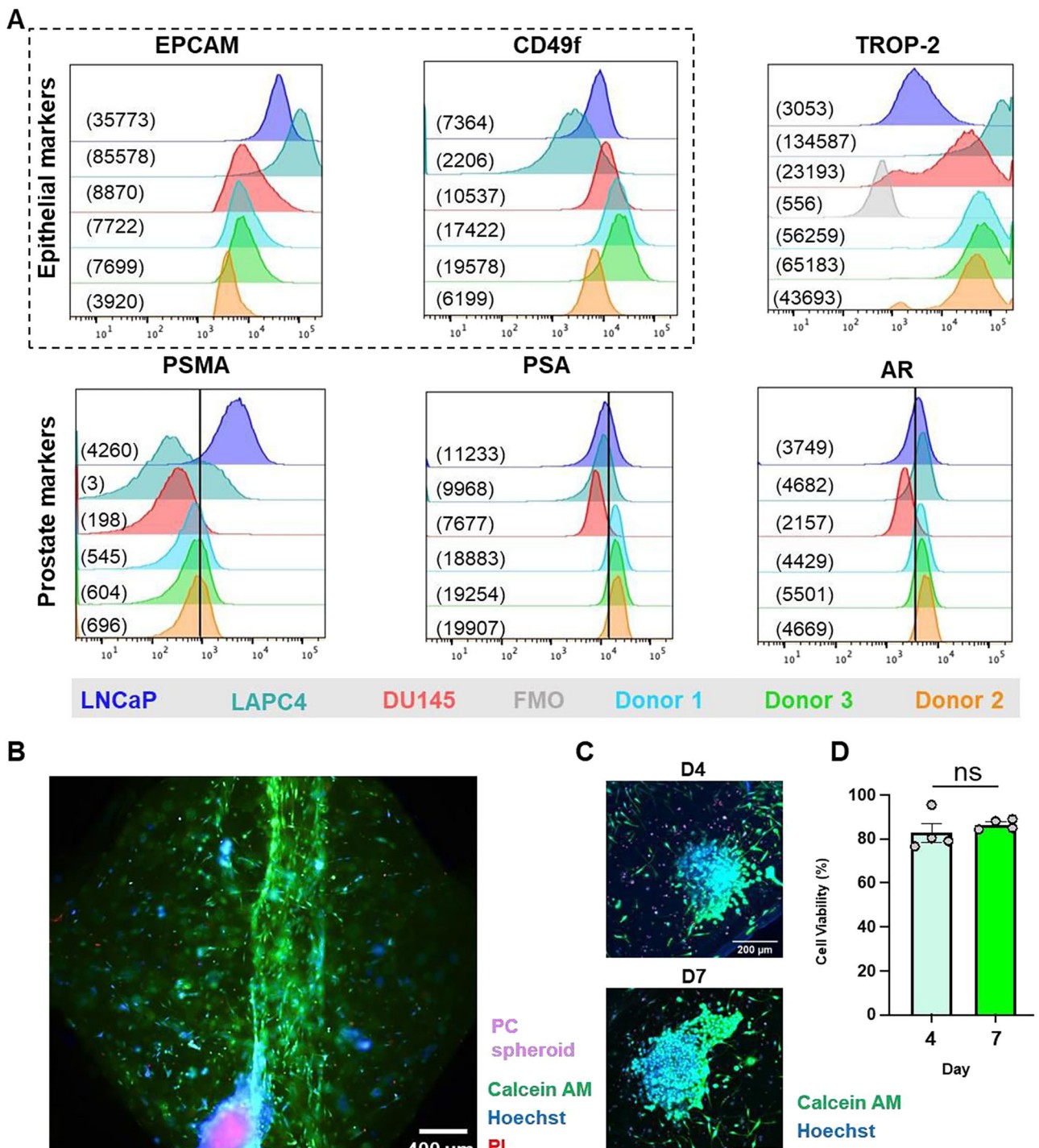

**Fig. 4 | Prostate cancer cell characterization. A** Flow cytometric analysis of epithelial and prostate cancer markers, with median fluorescence intensity (MFI) values for each condition shown in parentheses. TROP-2 FMO (Donor 1) served as negative control (gray) for TROP-2 staining. DU-145 staining served as a negative control for PCa biomarker staining, as known to be negative for these markers. Analysis was done in the live cell population. Gating strategy is provided in Supplementary Fig. 4. **B** Representative image of a MPS with a primary prostate epithelial cell spheroid. Cells were stained with calcein (green), PI (red), and Hoechst (blue), and tumor cells were stained with Cell Tracker (magenta). Scale bar: 400 μm. **C** Representative images and **D** quantification of cell viability of the prostate cancer spheroids at day 4 and 7 days of culture in the bone microenvironment MPS. Cells were stained with calcein (green), PI (red), and Hoechst (blue). Scale bars: 200 μm, N = 3 different donors. No statistical significance found through T-Test + Mann–Whitney test. Each point indicates a technical replicate. Figures show mean ± SD.

## Bone metastasis MPS predicts response to conventional treatments

To validate our MPS as a platform for reporting treatment responses in bone metastasis in prostate cancer, we initially created an MPS containing only spheroids of DU-145, LNCaP, and LAPC4 cell lines, which differ in AR status. LNCaP and LAPC4 are AR responsive, while DU-145 is resistant to AR therapy[71], allowing us to evaluate the system's response to AR-targeted and chemotherapy treatments commonly used to treat patients with

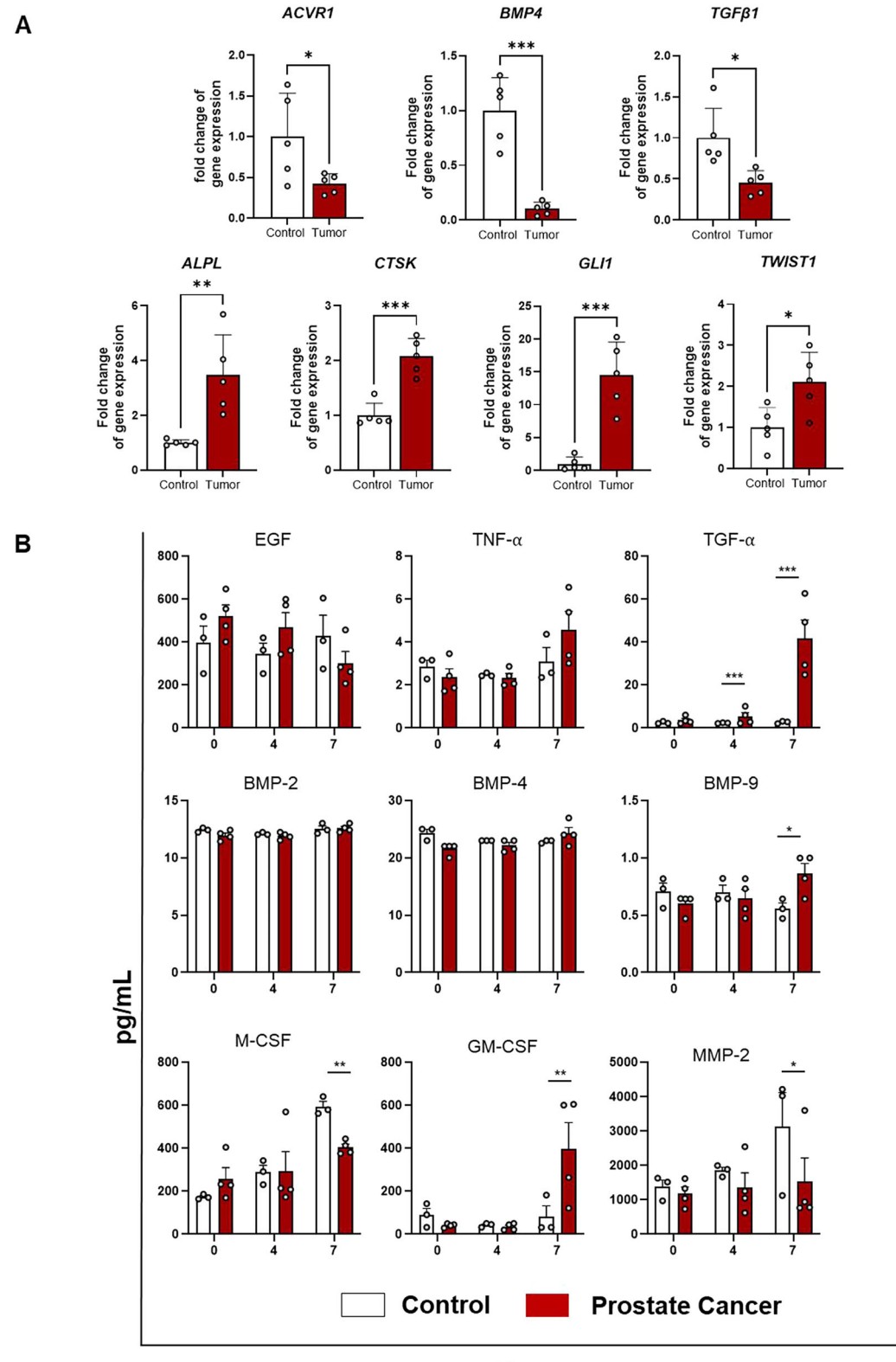

**Fig. 5 | Prostate cancer spheroids modify protein secretion in MPS. A** Bone metastasis MPS was assembled on day 0 with 10,000 stromal cells (MSCs, osteoblasts, adipocytes, macrophages, osteoclasts, fibroblasts) seeded in the main chamber. For prostate cancer devices, a spheroid was included in the port of the devices. After 4 days of co-culture, the main chamber was collected for RNA isolation and RT-qPCR analysis of stromal cells. Gene expression fold change of the significantly upregulated and downregulated genes found in the prostate cancer condition compared to the control. $n = 3$ replicates. Each point represents a technical replicate. *$p < 0.05$, **$p < 0.01$, and ***$p < 0.001$ (Student's $t$-test). **B** Media from the whole device was collected at day 0, 3, and 7, and soluble factors were analyzed with MAGPIX. $n = 4$. Each data point represents a technical replicate. *$p < 0.05$, **p < 0.01, and ***p < 0.001 (Student's $t$-test).

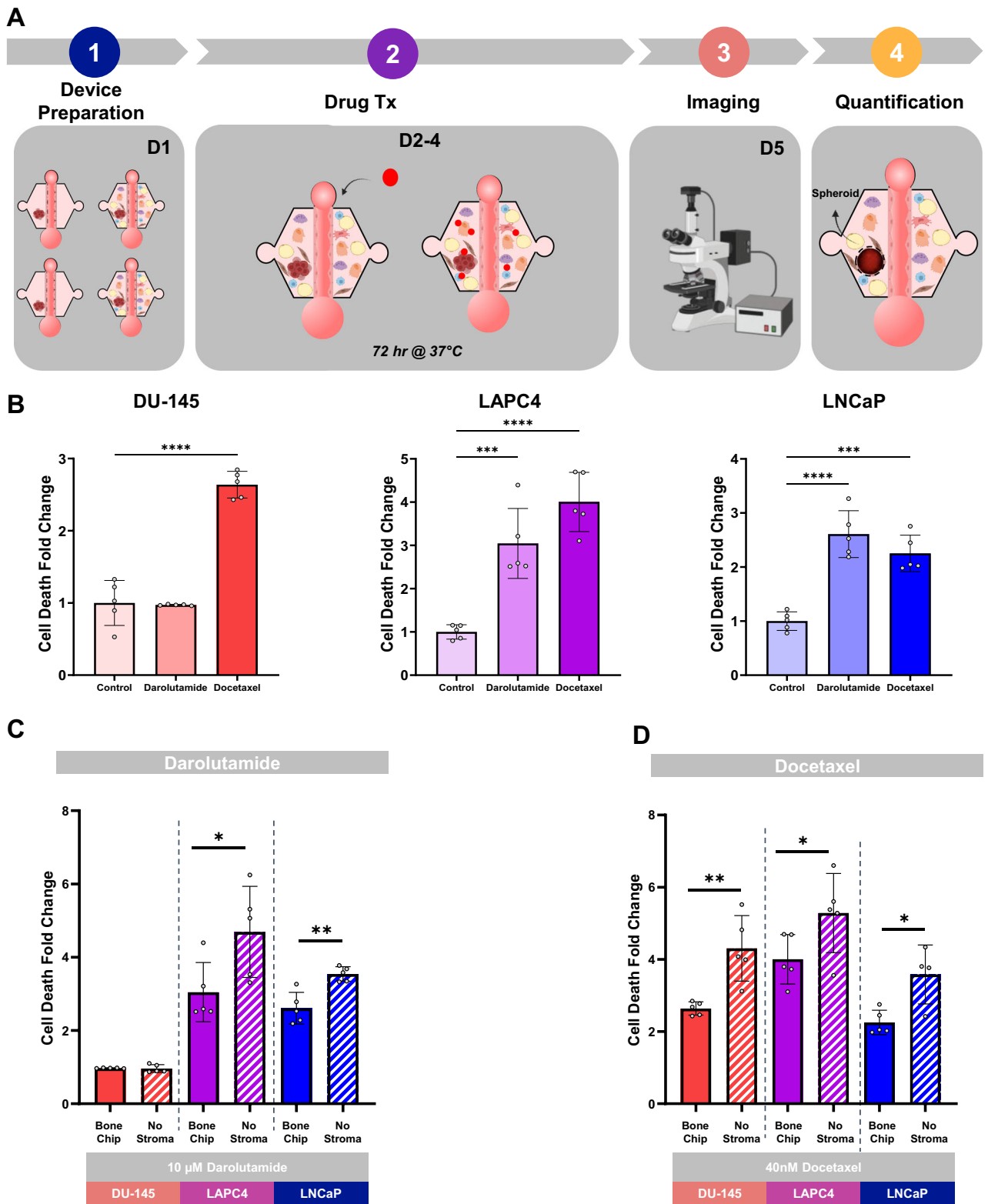

**Fig. 6 | Differential responses of androgen receptor-positive and -negative prostate cancer cells to treatment in a bone marrow microenvironment MPS.** **A** Schematic timeline of killing experiment in the bone metastasis MPS. The experimental timeline consists of five main stages: (1) initial spheroid formation of the MPS with DU-145, LAPC4, and LNCaP spheroids (2) treatment with either 10 µM darolutamide (AR antagonist) or 40 nM docetaxel (chemotherapy agent) over an additional 72-h period, (3) endpoint assessment via microscopy, and (4) data analysis of spheroid viability. **B** Prostate cancer cell line spheroid cell death across the three different cell lines measured as cell death fold change respective to the control treated with DMSO. $n = 5$. **C, D** Prostate cancer cell line spheroid cell death with (bone metastasis chip) and without embedded stromal cells. $n = 5$ Each point is a technical replicate. *$p < 0.05$ **$p < 0.01$, ****$p < 0.0001$ (A one way ANOVA, B Student's $t$-test). Figures show mean ± SD. Figure created with BioRender.com.

**Fig. 7 | TROP-2 ADC achieves significant killing of prostate cancer cells in 2D and 3D. A** DU-145 cells (TROP-2 medium) were seeded into a 96 well plate and treated with either TROP-2 ADC (SG) or Isotype ADC (5 μg/mL) for 1 h at 37 °C. Wells were subsequently washed three times to remove unbound ADC, and cell death fold change relative to the isotype ADC condition was quantified after 72 h. $n = 3$ replicates. Each point is a technical replicate. ****$p < 0.0001$ (Student's $t$-test). **B** LNCaP, DU-145, and LAPC4 prostate cancer spheroids were introduced into LumeNEXT devices without the bone stromal cells and treated with either the TROP-2 ADC or Isotype ADC (5 μg/mL) through the iPSC endothelial cell microvessel for 1 h at 37 °C. Devices were subsequently washed three times to remove unbound ADC, and cell death fold change relative to the isotype ADC condition was quantified after 72 h. $n = 3$ replicates. Each point represents a spheroid. **$p < 0.01$, ****$p < 0.0001$ (One-way ANOVA + Tukey's post-hoc test). Figures show mean ± SD. Figure created with BioRender.com.

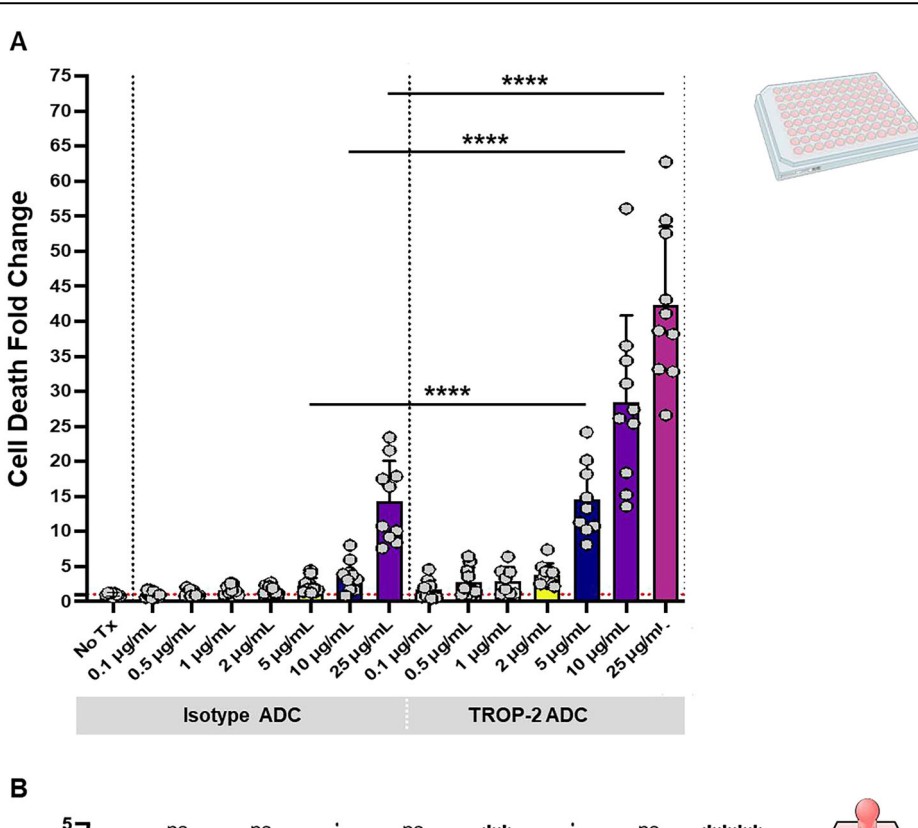

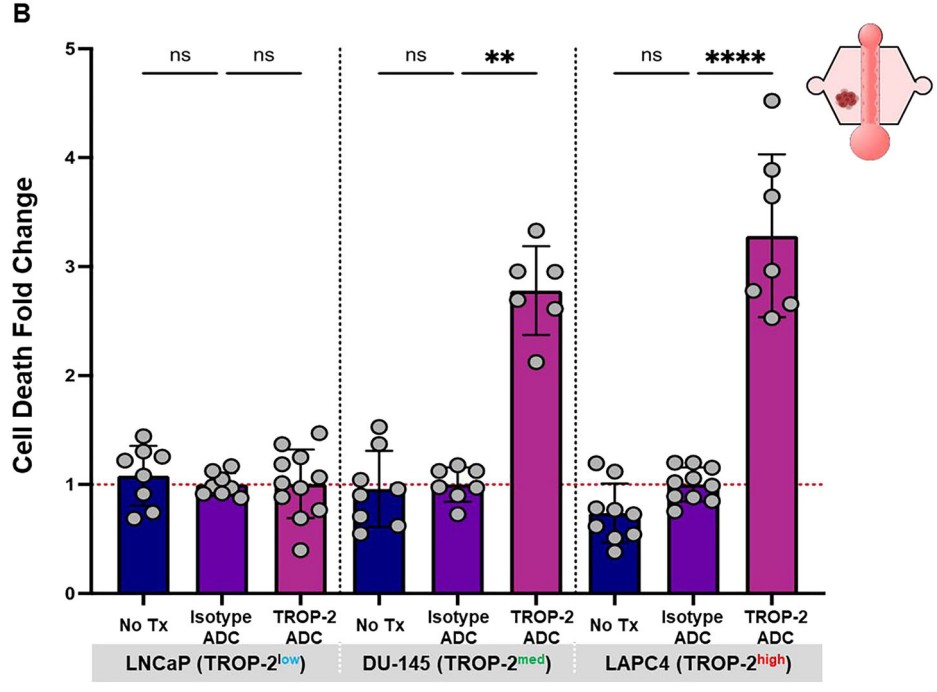

metastatic prostate cancer in the bone. The system was treated with 10 μM darolutamide, a second-generation AR antagonist designed to inhibit androgen signaling by preventing AR from binding to DNA and activating downstream growth-promoting genes. Darolutamide is specifically used in prostate cancer to block AR activity and reduce tumor proliferation, especially in cancers that retain sensitivity to androgens. In our model, treatment with darolutamide led to an increased cell death in the AR-positive LNCaP and LAPC4 spheroids (Fig. 6B). Conversely, the AR-negative DU-145 spheroids displayed minimal response to darolutamide, validating the specificity of AR-dependent effects within our model (Fig. 6B and Supplementary Fig. 6).

In addition to darolutamide, the spheroids were treated with 40 nM docetaxel, a microtubule-stabilizing chemotherapeutic agent that interferes

with cell division by preventing microtubule disassembly. Docetaxel is commonly used in advanced prostate cancer to induce cell cycle arrest and apoptosis, targeting rapidly dividing cancer cells regardless of AR status. In our MPS, docetaxel treatment resulted in pronounced cytotoxic effects across all spheroid types. DU-145, LNCaP, and LAPC4 spheroids exhibited substantial cytotoxic responses, suggesting a robust effect of Docetaxel on both AR-dependent and AR-independent cell lines (Fig. 6B and Supplementary Fig. 6).

Additionally, we evaluated the role of stromal cells within the bone metastasis MPS and their impact on the efficacy of darolutamide and docetaxel across the DU-145, LNCaP, and LAPC4 prostate cancer cell lines. We observed that the presence of stromal cells significantly decreased the treatment efficacy of both drugs. For AR-positive cell lines (LNCaP and

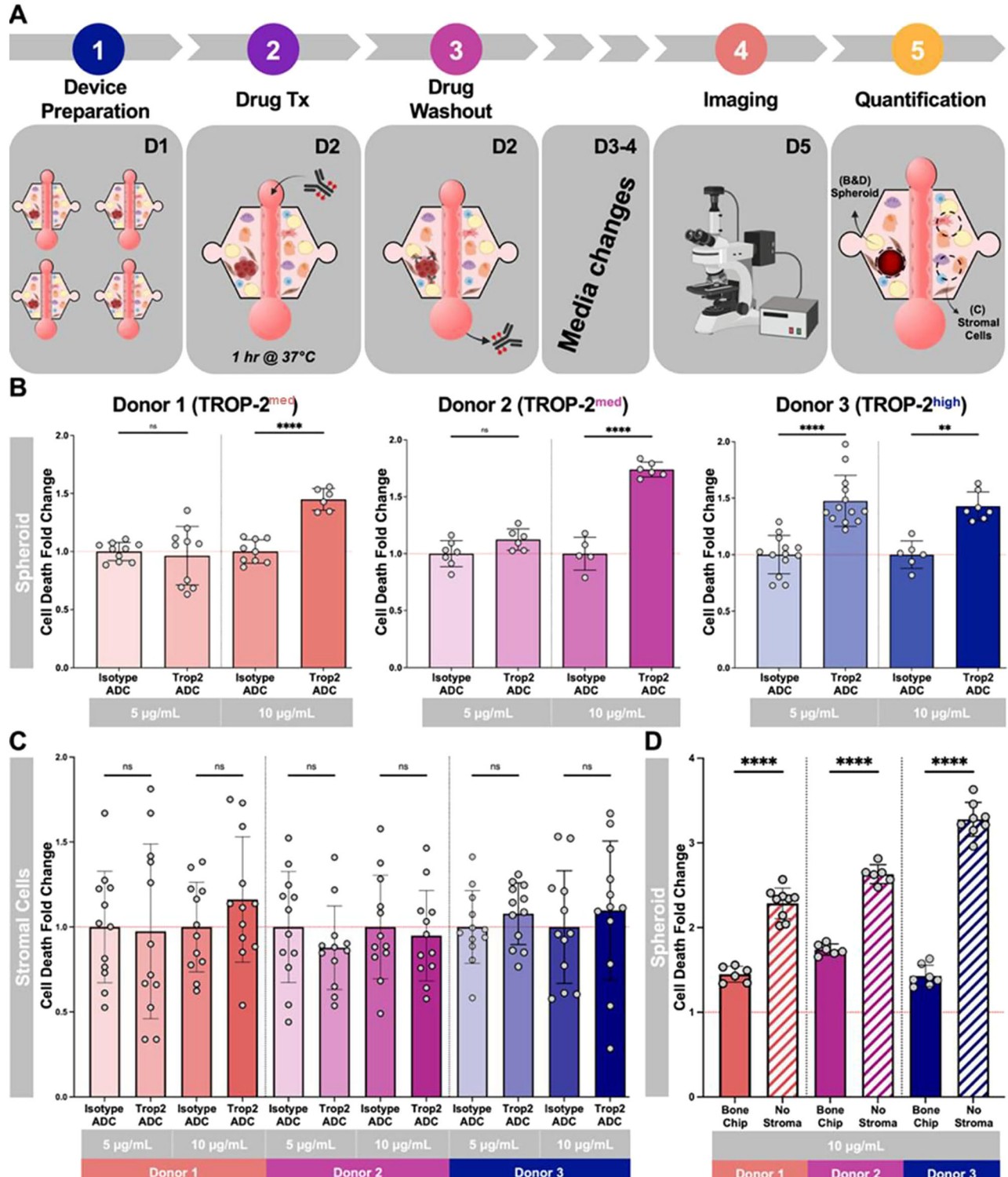

**Fig. 8 | Sacituzumab-Govitecan killing in the bone metastasis chip is donor dependent. A** Schematic timeline of killing experiment in the bone metastasis chip. **B** Primary prostate tumor epithelial spheroid and **C** stromal cell death across three donors with differing TROP-2 expression (medium, and high) at both 5 µg/mL and 10 µg/mL TROP-2 ADC (SG) measured as cell death fold change with respective to the isotype ADC control for each condition. **D** Primary prostate tumor epithelial spheroid cell death with (bone metastasis MPS) and without embedded stromal cells. $n = 3$. Each point is a technical replicate. ***$p < 0.005$, ****$p < 0.0001$ (Student's $t$-test). Figures show mean ± SD. Figure Created with BioRender.com.

LAPC4), stromal co-culture resulted in reduced sensitivity to darolutamide, as evidenced by lower cell death compared to spheroids treated without stromal cells (Fig. 6C). Similarly, docetaxel treatment was also less effective in the presence of stromal cells, with all three cell lines showing higher survival rates compared to their stromal-free counterparts (Fig. 6D). This

stromal-mediated protection against chemotherapy suggests that stromal cells in the bone marrow niche may contribute to a more resilient microenvironment, shielding cancer cells from cytotoxic agents (Fig. 6C, D and Supplementary Fig. 6). These findings suggest that stromal cells provide a protective microenvironment that can mitigate the inhibitory effects of

darolutamide and docetaxel, highlighting the role of the stroma in fostering therapeutic resistance.

## Sacituzumab govitecan has significant killing sensitivity and specificity for TROP-2+ cells

The development of new treatment strategies for metastatic PCa has been hampered by the lack of suitable models to evaluate treatment efficacy in a physiologically relevant microenvironment. Therefore, we next investigated the potential of the MPS for screening novel therapies such as ADCs. The design of the MPS mimics hematogenous delivery of therapies since drugs are added via the endothelial vessel and diffuse throughout the system, with a washout after treatment to mimic clearance (Supplementary Fig. 7).

Having examined responses to standard of care therapies in the MPS, we next investigated SG, an ADC currently undergoing a Phase II clinical trial in metastatic prostate cancer[8]. Initially, our goal was to identify optimal in vitro treatment conditions to evaluate SG's killing capacity while minimizing non-specific cell death. Given the limited availability of patient tumor tissue, we used a TROP-2 expressing prostate cancer cell line, DU-145, for initial characterization. We first optimized the drug dosing in 2D cultures due to their higher throughput, before dose testing in the 3D spheroid models. We treated DU-145 cells with SG and an isotype control ADC, which carries the SN-38 payload but does not get internalized, hydrolyze the linker, and does not release SN-38. The treatment was applied across a range of concentrations (0.1 µg/mL–25 µg/mL), based on values identified in the literature from both in vitro[72] and in vivo[73] studies (Fig. 7A). We quantified cell death and calculated the fold change relative to the no-treatment control. While the isotype ADC caused minimal cell death at lower concentrations, it spiked at the highest tested concentration (25 µg/mL) (Fig. 7A). In contrast, the SG treatment conditions demonstrated a dose-dependent response, with increasing cell death fold change with increasing concentration of drug (Fig. 7A). Our findings revealed that 5 µg/mL of SG was sufficient to achieve significant killing compared to the isotype ADC, while non-specific cell death was minimal under the respective isotype ADC condition.

Subsequently, we aimed to evaluate the selected concentration in a 3D microenvironment. To achieve this, we generated spheroids from the LNCaP, DU-145, and LAPC4 prostate cancer cell lines, with varying levels of surface TROP-2 expression (low, medium, and high, respectively) (Fig. 7B) and incorporated them into the hydrogel of a LumeNEXT device, as described previously. The lumen of the device was seeded with iPSC ECs for microvessel formation, which served as the route of drug delivery for subsequent studies. For initial experiments, spheroids were cultured in the collagen matrix without stromal cells. Similar to the 2D assay, spheroids in the LumeNEXT platform were treated with either no treatment, SG, or the isotype ADC at 5 µg/mL for 1 h at 37 °C. Cell death was visualized using PI staining, and a z-stack image was captured for each device to capture signal from the entire spheroid. To compare across experimental replicates and between experimental conditions, the fold change in cell death was also determined for each condition, using the isotype ADC condition as a reference point. In the TROP-2[low] (LNCaP) spheroids there was no significant increase in cell death in the SG condition compared to the isotype ADC (Fig. 7B). However, both TROP-2[med] (DU-145) and TROP-2[high] (LAPC4) spheroids showed a robust increase in cell death fold change when treated with SG (Fig. 7B). These observations highlight the variations in cell killing efficacy dependent on TROP-2 expression levels.

Finally, we assembled the bone metastasis MPS with primary prostate tumor epithelial spheroids from three unique patients with varying and distinct levels of TROP-2 expression (Fig. 4), alongside the endothelial vessel, and the stromal compartment. Notably, while all of the cells in the MPS are primary cells, matched bone marrow aspirates were not available, therefore, the cells comprising the stromal compartment and primary prostate tumor epithelial spheroids were not from the same patient. However, this does allow us to determine patient tumor responses dependent on TROP-2 levels, as the bone stromal microenvironment

remained constant between the donors. Donor 1 and Donor 2 were moderate expressors, while Donor 3 had the highest TROP-2 expression. We treated the MPS with 5 µg/mL and 10 µg/mL SG. Treatment with 10 µg/mL of SG in the Donor 1 and Donor 2 device achieved a significant increase in cell death fold change while only 5 µg/mL was necessary for Donor 3 (Fig. 8A, B and Supplementary Fig. 8). In parallel, the viability of the stromal cells was measured in each treatment condition for all three patient MPS. Interestingly, the stromal compartment had minimal cell death overall and was not significantly affected by SG treatment compared to the isotype ADC treatment (Fig. 8C and Supplementary Fig. 8). We also compared the killing efficacy of SG on the primary prostate tumor epithelial spheroids with and without the stromal compartment present in the MPS. Cell death was significantly higher in all three patient spheroids in the absence of stroma, suggesting the stromal cells attenuate the efficacy of SG (Fig. 8D). Overall, these studies highlight that in our MPS, we observe TROP-2 specific targeting of SG with minimal non-specific killing in optimized treatment conditions.

## Discussion

Bone metastases are associated with significant morbidity and mortality in many cancer types, especially prostate cancer. Patients with bone metastatic disease often exhibit reduced responsiveness to conventional therapies, resulting in a poor prognosis, and this poses a challenge to deliver effective treatment. Therefore, gaining a comprehensive understanding of how the bone metastatic tumor microenvironment is influenced by prostate cancer, and vice versa, is imperative.

Currently used models for studying PC, such as in vitro 2D cultures, have several limitations. Often, these systems fail to capture the complex cell-to-cell signaling that is a cornerstone of the bone niche. Mouse models represent one alternative approach to model the complex bone microenvironment, but have had limited clinical translational potential due to the differences between rodent and human bone physiology, as well as the requirement for immunodeficient mice for human xenograft models. Additionally, animal models can be costly, time-consuming, restricting their use for high-throughput studies. In this context, humanized mouse models[38] have been used to provide insights into human bone cell interactions; however, our MPS offers significant advantages for studying PCa bone metastases. Unlike in vivo models, our fully human-based system eliminates species-specific limitations and allows for precise control over the microenvironment. It enables real-time monitoring of dynamic processes like tumor-stroma interactions and therapy responses while supporting patient-specific modeling for a more personalized approach. However, the MPS system presented in this study does lack the full complexity of the bone marrow niche, including rare hematopoietic populations and long-term immune adaptation. Additionally, inter-patient heterogeneity of primary cells can introduce inconsistencies, requiring increased sample sizes for reproducibility while enhancing personalized medicine applications.

Thus, there is a pressing need for improved models that overcome the limitations of animal models. More advanced in vitro approaches, including hydrogel-based cultures, organ-on-a-chip platforms, and bioreactor systems, offer improved biomimicry but often come with limitations in terms of scalability and the ability to incorporate multiple cell types[11,38,40,42,74]. Many bone-mimetic platforms lack cellular diversity, immune cells, or rely on murine stromal components or immortalized cell lines. Our MPS overcomes these challenges by integrating a diverse set of human-derived cells, allowing for more accurate representation of metastatic prostate cancer progression in the bone microenvironment. Using the LumeNEXT platform, we successfully developed an MPS that incorporates six critical stromal cell types found in the bone microenvironment. The spheroids varied in size, which mirrors the natural variability observed in tumors, further enhancing the model's relevance to in vivo conditions. Furthermore, we studied primary prostate tumor epithelial spheroids from three different patients, allowing for the investigation of patient-specific responses to better reflect the heterogeneity of disease. This use of spheroids further underscores the innovation and translational relevance of this work, providing a

more accurate and personalized model for studying prostate cancer bone metastasis and testing therapeutic strategies. In addition to the primary prostate tumor epithelial spheroids and bone stromal cells, we included iPSC-derived endothelial cells to form a microvessel to recapitulate the endothelium of the TME, as bone is highly vascularized. This model represents one of the most complex biological 3D co-culture bone MPS developed to date.

We successfully cultivated both stromal cells and primary prostate tumor epithelial spheroids in our MPS for 7 days, which was sufficient to provide readouts of treatment efficacy. This also establishes a foundation for future experiments with extended culture times that would support study of long-term drug effects, chronic disease progression, or sequential treatment testing, which requires longer culture times. We observed continuous evolution of the system over time as stromal cells continued to differentiate throughout the experimental timeframe. Notably, we observed an increase in lipid droplet deposition, along with enhanced expression of markers for adipocytes, osteoblasts, fibroblasts, and osteoclasts as the experiment progressed. In addition to the comprehensive characterization of our platform, we validated our MPS as a model for studying bone metastasis in prostate cancer, where we created a 3D culture with prostate cancer spheroids (DU-145, LNCaP, and LAPC4) representing different AR statuses. LNCaP and LAPC4 are AR-positive, while DU-145 is AR-negative, enabling us to evaluate both AR-targeted and general chemotherapy responses. Our device effectively captured the sensitivity of these spheroids to AR inhibition and chemotherapy treatments.

Our study highlights the potential of the MPS as a robust platform for evaluating novel therapeutics, including ADCs like SG, in a biologically relevant model of metastatic prostate cancer in the bone niche. We found that the response to SG treatment was dose-dependent and influenced by TROP-2 expression levels, demonstrating the platform's ability to discern differential drug sensitivities (Fig. 8). While lower TROP-2-expressing spheroids required higher SG concentrations for efficacy (Donor 1 and Donor 2), higher TROP-2-expressing spheroids (Donor 3) responded at lower doses, underscoring the potential for this system to inform patient stratification and optimize treatment regimens while minimizing toxicity. Furthermore, the ability of both docetaxel and SG to induce tumor cell killing supports the MPS's capacity to sustain prostate cancer spheroid proliferation, as this is required for the mechanism of action for these drugs, though further studies on proliferation are warranted. Importantly, the reduced efficacy of all tested treatments in the presence of stroma demonstrates the platform's ability to recapitulate treatment resistance mechanisms inherent to the tumor microenvironment. This feature positions the MPS as a powerful tool for uncovering the cellular and molecular drivers of resistance. This is an area of interest for future work to permit identification of the cell types and mechanisms that drive treatment resistance.

Further studies will investigate in depth the mechanisms underlying chemotherapy resistance in the bone microenvironment. One of the primary mechanisms that might be responsible for reducing chemotherapy efficacy in the bone is alterations in ECM, including changes in collagen deposition and increased bone mineralization, which creates a physical and biochemical barrier that limits drug penetration[52,75,76]. Additionally, cancer cells within the bone microenvironment can exploit signaling pathways that enhance survival and promote drug resistance. We observed that the presence of tumor cells upregulated the expression of TGF-α and EGF, both of which are abundantly released during osteoclastic bone resorption (Fig. 5)[77]. TGF-α plays a crucial role in stimulating tumor growth and enhancing resistance to therapy by activating the EGFR pathway[78]. This leads to downstream activation of survival signaling cascades, including MAPK/ERK and PI3K/Akt, which promote tumor proliferation and inhibit apoptosis[78,79]. Furthermore, TGF-α and EGF signaling drive epithelial-to-mesenchymal transition (EMT), a process that enhances tumor cell motility, invasion, and resistance to chemotherapy. In MPS containing prostate tumor spheroids, we observed upregulation of EMT-related transcription factors GLI and TWIST1, compared to MPS without tumors. These results further reinforce a mesenchymal, drug-resistant phenotype. GLI, a key

effector of Hedgehog signaling, has been implicated in maintaining cancer stem-like properties, while TWIST1 plays a crucial role in EMT induction and resistance to apoptosis (Fig. 5)[80,81]. Activation of these pathways confers resistance to chemotherapy by promoting stemness, increasing DNA repair capacity, and suppressing pro-apoptotic signaling[82].

A limitation of this study was the use of primary prostate tumor epithelial spheroids and MSCs obtained from different patient donors, as acquisition of donor-matched specimens was not logistically feasible for this study. However, this is an area of interest for future MPS development to use matched bone marrow aspirates, tumor epithelial cells, and monocytes from a single patient. Further, the cell populations added into the device were derived from MSCs, therefore, they could still include some undifferentiated MSCs. Osteoclasts were also differentiated from peripheral blood monocytes from patients with prostate cancer. Variability in differentiation rates across patient-derived samples likely contributes to differences in final osteoclast numbers, as the differentiated population contains mainly osteoclasts with some M2 macrophages. We observed that the proportion of osteoclasts inside the devices ranged from 2% to 9% at days 4 and 9. This variation could also be influenced by the continuous differentiation of M2 macrophages towards the osteoclast lineage, as suggested by the increasing expression of osteoclast-related genes over time, as measured by qPCR (Supplementary Fig. 2). This could be addressed in future studies by developing sorting techniques to isolate pure populations of these cell types either from bone marrow aspirates or differentiated cells. However, at the time of performing these experiments, there were no published protocols for separating bone cell populations suitable for this purpose. For standardization purposes, we used the same relative percentages of the different cell types in all devices, despite the significant variability in the bone niche composition due to factors such as age, medication usage, and underlying pathological conditions. scRNA-seq was performed to confirm and quantify cell types in the device based on gene transcription. However, scRNA-seq analysis revealed challenges in detecting certain cell populations. For example, low transcriptional reads were noted for osteoclasts and adipocytes, likely due to their small numbers and transcriptional heterogeneity as they differentiated from MSCs, respectively. The qPCR data further supports this differentiation process, showing a reduction in MSC gene expression, indicating that the cells are not proliferating but continuing to mature into specialized cell types (Supplementary Fig. 2). Osteoclasts, comprising less than 10% of the population, were present in low numbers, resulting in insufficient scRNA-seq detection, though qPCR was able to detect osteoclast-specific genes, including CTSK and ATP5, thanks to its higher sensitivity for low transcriptional reads. Similarly, adipocytes presented an additional challenge, as they were observed only in minimal amounts. Adipocytes are challenging to capture in scRNA-seq due to their large size, low RNA yield, and lipid-rich cytoplasm, which hinder efficient RNA extraction[69,83].

Finally, while TROP-2 expression levels were initially hypothesized to be the main driver of prostate tumor epithelial spheroid response to varying SG dosages, our model reveals that other components of the TME may significantly influence the response to therapy. Further characterization of cancer cell response to therapy through depletion of different stromal elements could help to shed light on the impacts of the stromal compartment on tumor viability.

In conclusion, we have successfully developed a complex MPS that more closely recapitulates the phenotype of PCa metastases within the bone niche. This platform represents a significant advancement in our ability to understand the intricate crosstalk between prostate tumor epithelial cells and the tumor microenvironment. The MPS provides a robust platform to permit the evaluation of multiple different therapeutics, including novel therapies, and demonstrates the potential for use of the model for preclinical testing. Furthermore, this MPS holds great promise for personalized medicine, as it enables the incorporation of patient-derived cells to better reflect individual tumor heterogeneity and treatment responses[52]. By integrating patient-specific stromal cells, monocyte-derived osteoclasts, and tumor epithelial spheroids, this system can be tailored to model the unique

microenvironment of each patient's bone metastases. The use of fully matched models, including stromal cells, could yield different responses to therapeutics. Such an approach could improve precision oncology by facilitating the testing of multiple therapeutic strategies on a patient-by-patient basis and help identify the most effective treatment options while minimizing toxicity. Future refinements of the model, including incorporation of patient-matched samples, will allow us to investigate how stromal heterogeneity influences tumor progression and therapy resistance. Thus, this MPS will serve as a building block for developing fully matched patient-specific models of PCa bone metastasis, allowing for a more personalized approach to studying the role of distinct cell types in treatment response and resistance.

## Methods

### Collection of primary tissue specimens and cell isolation

All patients provided written, informed consent under an Institutional Review Board (IRB) approved protocol at the University of Wisconsin-Madison. All ethical regulations relevant to human research participants were followed. Bone marrow-derived mesenchymal stem cells were isolated from bone marrow aspirates collected from the pubic symphysis of consenting prostate cancer patients undergoing radical prostatectomy. Mononuclear cells were isolated by gradient centrifugation separation on Lymphocyte Separation Media (Corning). Cells were plated at a density of $1–4 \times 10^5$ cells/cm$^2$, in alpha-MEM containing 15% human platelet lysate (hPL, StemCell Technologies), 2 IU/mL heparin (StemCell Technologies), 1% Glutamax, and 1% penicillin/streptomycin. Non-adherent cells were removed 72 h later, leaving adherent MSCs. MSCs were then expanded, maintained in 5% hPL containing media, and used up to passage 8. At passage 3, standard MSC markers were evaluated by flow cytometry using CD105, CD73, and CD90, after gating on live, single cells negative for immune and endothelial markers. MSCs were also evaluated for the ability to differentiate into osteoblasts and adipocytes. The above MSC identification criteria were performed according to guidelines from the MSC committee of the International Stem Cell Society[84].

Human prostate tissues were obtained at the University of Wisconsin-Madison from patients with prostate cancer undergoing radical prostatectomy who had received no prior treatments. The University of Wisconsin IRB approved utilization of all the tissue samples in this study, and written and informed consents were obtained from all patients. For the primary patient-derived prostate tissue, the prostate cancer diagnosis was confirmed by a pathologist for all samples. Tissue was collected from prostatectomy samples under an approved IRB protocol and was maintained in transport media (DMEM basal media with gentamycin, amphotericin, and penicillin/streptomycin (Penicillin/Streptomycin) at 1% v/v each) prior to processing. Samples were then minced and transferred to a 15 mL conical tube containing digestion media (6 mL of transport media with 0.5% collagenase (Thermo-Fisher, 17100017), 0.1% Dispase (Worthington, LS02100), and 0.1% DNAse (Roche, 04716728001). The digestion mixture was incubated for 4 hours at 37 °C with rotation. To isolate cells from the sample, the sample was washed with Prostate Epithelial Cell media (PEpiCM, ScienCell). Then, the digested sample was filtered using a 40 μm cap filter and centrifuged at $400 \times g$ for 10 min[85]. The cell pellet was washed twice with PBS and then cultured in Prostate Epithelial Cell media (PEpiCM, ScienCell).

For monocytes, peripheral blood from prostate cancer donors was collected and monocytes were isolated using RosetteSep™ Monocyte Enrichment Cocktail (Stemcell) and following the manufacturer protocol. Briefly, RosetteSep™ Enrichment Cocktail was added to whole blood and incubated for 10 min. The blood was diluted isolated by gradient centrifugation separation 1:1 in PBS + 2% FBS. Monocytes were isolated by gradient centrifugation separation, using SepMate tubes and Lymphoprep. Tubes were centrifuged for 10 min at $1200 \times g$ at room temperature with brake on. Enriched cells were washed with PBS and plate at a density of $10^6$ cells/cm$^2$ in monocyte attachment media (Promocell) for 1 h.

### Cell culture and differentiation

Human MSCs were maintained in culture media (DMEM, 10% FBS, 1% Glutamax, and 1% pen/strep). For differentiation, MSCs were plated at 20,000 cells/cm$^2$. For adipocyte differentiation, MSCs were differentiated for 3 weeks in MesenCult™ Basal Media supplemented with MesenCult™ Adipogenic Differentiation supplements (StemCell). For osteoblast differentiation, MSCs were differentiated for 2 weeks in MesenCult™ Osteogenic Differentiation Kit (StemCell). For fibroblast differentiation, MSCs were differentiated in MesenCult Basal Media with 1% P/S and 2 mM Glutamine, 100 ng/ml CTGF, and 0.25 μg/ml Ascorbic Acid. For osteoclast differentiation, monocytes were cultured in DMEM 10% FBS with 100 ng/ml m-CSF for 3 days to induce macrophage polarization. Then, media was changed, and macrophages were cultured in DMEM with 100 ng/ml m-CSF and 50 ng/mL RANKL for 12 days. Primary prostate cells were cultured in PEpiCM (ScienCell) supplemented with PEpiCGS (ScienCell), 1% Glutamax, and 1% pen/strep. iPSC cells were cultured in a standard T75 cm2 cell culture flask coated with fibronectin. iPSC cell cultures were maintained in Vasculife VEGF Endothelial Medium (LifeLine) supplemented with iCell Endothelial Cells Medium Supplement (Cellular Dynamics), 5 ng/ml rh FGF basic factor, 50 μg/ml Ascorbic Acid, 1 μg/ml Hydrocortisone Hemisuccinate, 15 ng/ml rh IGF-1 factor, 5 ng/ml rh EGF factor, 5 ng/ml rh VEGF factor, 0.75 U/mL Heparin Sulfate, 30 mg/ml gentamicin, and 15 μg/ml Amphotericin B.

### Prostate cancer cell line spheroids

DU-145, LNCaP, and LAPC4 were obtained from ATCC. DU-145, LNCaP, and LAPC4 spheroids were generated using a modified hanging drop protocol supplemented with ECM gel (Sigma). Cells were trypsinized and resuspended at 50 cells/μL and mixed in equal volume with ECM gel. 50 μL droplets were dispensed on the surface of a 24-well plate. The plate is subsequently placed in the incubator for 2 min allow for the ECM gel to begin polymerization. The plate was then inverted and placed in the incubator for 10–15 min. After the ECM gel was polymerized, the plate was placed right side up, and 500 μL of media was added to each well. Droplets were cultured for 9–10 days before spheroids were harvested for use in experiments.

### Primary prostate tumor epithelial spheroids

Primary prostate tumor epithelial spheroids were generated by the hanging drop method described previously[86,87]. Briefly, cells were trypsinized, counted, and resuspended at 6000 cells/μl in media supplemented with 20% 12 g/l methylcellulose. Twenty-five microliters of droplets were placed on top of a Petri dish lid, and distilled water was added to the bottom of the dish to maintain humidity during spheroid formation. After a 48 h incubation, one single spheroid per droplet was formed.

### Media optimization

MicroDuo cell culture plates were sterilized and fibronectin-coated. Primary prostate cells, iPSC ECs, and macrophages were trypsinized and seeded in 12.5 μl/well of its own media. After 24 h, media was aspirated and replaced by 25 μl/well of the different media combinations. Cells were cultured at 37 °C for 5 days. At day 6, media was replaced with 12.5 μl/well of Cell-Titer Glo. The plate was incubated for 15 min, and then luminescence was read. The different media formulations are described in Supplemental Table 1.

### Device fabrication

LumeNEXT was fabricated in PDMS using standard soft lithography techniques as previously described in ref. 53.

Briefly, polydimethylsiloxane (PDMS, Dow Corning, Sylgard 184) was mixed at a 10:1 ratio and was poured over SU-8 silicon master molds. 25-gauge hypodermic needles were filled with the mix to fabricate the rods. PDMS components were then baked at 80 °C for 4 h. After baking, two layers of the LumeNEXT device were aligned, and the PDMS rods were extracted from the needles and sandwiched between them. The devices were

oxygen plasma bonded to a glass-bottom MatTek dish (MatTek Corporation, P50G-1.5-30-F) and UV sterilized for 15–20 min before use.

## Preparation of bone microphysiological systems

To maximize hydrogel adhesion chamber, devices were treated with 2% poly(ethyleneimine) for 10 min followed by a 0.4% glutaraldehyde treatment for 30 min. Finally, the microdevices were washed four times with sterile deionized water. High-density rat-tail collagen type 1 (Corning, referred to as collagen through the text) was diluted with 10x PBS and neutralized with 0.5 M NaOH (Fisher Scientific) for a final concentration of 1x PBS, and a pH of 7.4. 5ug/ml fibronectin and 1 mg/ml fibrinogen were added to the mixture. A final concentration of 4 mg/mL collagen type I was achieved by adding DMEM containing the cell suspension mix. For the cell suspension, osteoblasts, adipocytes, fibroblasts, MSCs, macrophages, and osteoclast cultures were trypsinized and counted. For every 100 µl of cell suspension we mixed 2500 fibroblasts, 100,000 osteoblasts, 75,000 adipocytes, 25,000 MSCs, 47,500 M2 macrophages, and 47,500 osteoclasts. The cell mix was centrifuged and resuspended in MCM (Supplementary Table 1). Then, 6 µl of the collagen solution was mixed with spheroid solution and loaded through the side ports and polymerized at room temperature for 10 min. For gene expression analysis, spheroids were mixed with collagen solution and placed in the side port. Devices were transferred to 37 °C for 1 h to allow collagen to polymerize fully.

After incubation, a droplet of media (5 µL) was added to the input port. Then, the rod was pulled through the output port using sterilized tweezers, spontaneously filling the tubular cast with media. The microdevice design's passive pumping mechanism transports media through the channel[53]. iPSC ECs were trypsinized and resuspended in multi-co-culture media at 20,000 cells/µL and seeded into the lumens (3 µL per lumen). Lumens were incubated at 37 °C for 2 h to allow for cell attachment. Then, lumens were washed 3 times with 10 µL of multi-co-culture media added per lumen and incubated at 37 °C. Media in the lumens was refreshed twice daily.

## Single-cell sequencing

The top PDMS layer of the bone MPS was removed, and the hydrogel removed with tweezers into an 8 mg/ml collagenase I (Worthington) solution in PBS. 18 MPS were pooled per sample. The matrix was digested for 5 min at 37 °C prior to centrifugation to pellet the cells, and washed with PBS to remove the enzyme solution. Cells were then fixed with Fixation Buffer (PN-20000517, 10X Genomics) overnight at 4 °C. Cells were then pelleted, and fixation buffer was removed and replaced with Quenching Buffer (PN-20000516, 10X Genomics). Cells were quantified using a Countess (Thermo) and stored at 4 °C with Enhancer added (PN-20000482, 10X Genomics) to allow batch processing of the samples. Probe hybridization, GEM generation, and barcoding were performed using the Chromium Fixed RNA kit, Human transcriptome (PN-1000475, 10X Genomics), following the manufacturer's instructions. GEM generation was performed using a Chromium Next GEM chip Q single cell kit (PN-1000418, 10X Genomics) on a Chromium X (10X Genomics). Samples were pre-amplified using 7 cycles of PCR, and cDNA libraries were constructed as detailed in the manufacturer's protocol. cDNA libraries were submitted for next-generation sequencing at the University of Wisconsin-Madison Gene Expression Center at a depth of 70,000 reads per cell on a NovaSeq X Plus.

Samples from days 4 and 7 were initially preprocessed using CellRanger. All samples were aligned with Human (GRCh38) 2020-A reference genome, which was available through the 10x Genomics cloud platform. Quality control steps, performed using Seurat v5, removed cells with over 5% mitochondrial reads to filter damaged cells or those with high reactive oxygen species (ROS) content. The Day 4 and Day 7 datasets were loaded as HDF5 files into Seurat, where quality control was applied independently to each dataset prior to integration. The Harmony method was used to perform batch correction and integration. Following integration, clusters were identified using Seurat, which initially resulted in 10 clusters with resolution set to 0.35. Cluster annotation was conducted manually using canonical markers from literature and PanglaoDB, supported by differential gene expression analysis to distinguish clusters from each other.

The cell type identification utilized specific markers to distinguish between cell types in the dataset. Endothelial cells were identified by VWF, while COL1A1 served as the marker for osteoblasts, and ADIPOQ indicated adipocytes. MSCs were marked by *ADAMTS4*, *TGM2*, and CD9, and SRGN was used to identify M2 macrophages. Fibroblasts were recognized by their expression of COL4A1. These markers provided a framework for defining each cell type within the clusters, aiding in the differentiation and classification of cells in various stages of their lineage.

## Darolutamide and docetaxel 3D killing assay

DU-145, LNCaP, LAPC4, and primary prostate spheroids were generated as described above. Spheroids were added into the ECM mixture (±stromal cells) and introduced into the LumeNEXT devices through the side port. Lumen formation was performed as described above. After 24 h, the devices were treated with DMSO (control), 10 µl of 10 µM darolutamide, or 40 nM docetaxel for 72 h. Then, the spheroids were stained with Hoechst (1:1000) and propidium iodide (1:1000). Cells were imaged on a fluorescent microscope, wherein a z-stack was collected for each spheroid. Using ImageJ, a maximum intensity projection for each spheroid was generated, and a ROI was drawn around the perimeter of the spheroid using the Hoechst staining for reference. The mean intensity value of the PI signal was measured inside the ROI for each spheroid. Cell death fold change with respect to the DMSO control for each cell type and treatment concentration was calculated.

## Sacituzumab govitecan killing assays

Sacituzumab govitecan (SG) and isotype ADC were provided by Gilead Sciences.

Initial optimization of SG treatment parameters was performed in 2D using TROP-2+, DU-145 prostate cancer cells. DU-145 cells were seeded in a 96-well plate at 5000 cells/well. After 24 h, cells were treated with a range of concentrations (0.1 µg/mL – 25 µg/mL) of both SG and the Isotype ADC or received media only. Cells were placed in an incubator for 1 h and were subsequently washed with fresh media three times to remove unbound drug. After 72 h, the DU-145 cells were stained with Hoechst (1:1000) and propidium iodide (1:1000). Cells were imaged on a fluorescent microscope, and the fold change in cell death with respect to the no-treatment control was calculated after analysis on ImageJ.

For the 3D killing assays, DU-145, LNCaP, LAPC4, and primary prostate spheroids were generated as described above. Spheroids were added into the ECM mixture (±stromal cells) and introduced into the LumeNEXT devices through the side port. Lumen formation was performed as described above. After 24 h, SG and the isotype ADC were reconstituted in the multi-phenotype media at the desired concentrations (5 and 10 µg/mL). SG was added through the lumen inlet port, and devices were incubated for 1 h at 37 °C. Devices were subsequently submerged in a 100 µL droplet of MCM for 30 min, three times, to remove unbound ADC. After washing, devices were replenished with 10–15 µL of MCM through the lumen. Media was changed daily, and after 72 h, the spheroids were stained with Hoechst (1:1000) and propidium iodide (1:1000). Cells were imaged on a fluorescent microscope, wherein a z-stack was collected for each spheroid. Using ImageJ, a maximum intensity projection for each spheroid was generated, and a ROI was drawn around the perimeter of the spheroid using the Hoechst staining for reference. The mean intensity value of the PI signal was measured inside the ROI for each spheroid. Cell death fold change with respect to the isotype ADC for each cell type and treatment concentration was calculated.

## Cell viability

Cell viability in stroma and spheroids was evaluated by confocal microscopy. Devices were stained with Calcein AM, propidium iodide (PI), and Hoechst. Briefly, Calcein was diluted 1:4, PI and Hoechst were diluted 1:1000 in serum-free media. Staining mix was added in the input port of the

device and incubated at 37 °C for 30 min. Devices were washed 3x with MCM and imaged using a Leica SP8 STED confocal microscope.

## Flow cytometry

Cells were stained with 1:100 Ghost Dye™ Violet 510 fixable live/dead stain (Tonbo Biosciences, San Diego, CA), 1:100 Fc blocker (Fc Block, BD Biosciences), and fluorescently labeled antibodies, including 1:100 EpCAM Brilliant Violet 650, 1:100 TROP2 PE, 1:100 CD49f PerCP Cy5.5, 1:100 PSMA PE-Cy7 (Biolegend). For intracellular staining, fixation and permeabilization were performed following the manufacturer's protocol (eBioscience, Thermo Fisher Scientific, MA), followed by staining with 1:100 PSA-Cy5 (Bioss), 1:200 AR (Cell Signaling), and 1:200 Donkey Anti-Rabbit Alexa Fluor488 (Biolegend). Cells were acquired on a BD LSRII instrument (BD Biosciences, Franklin Lakes, NJ, USA). Data was analyzed with FlowJo v10.7.1 (FlowJo LLC, by BD Biosciences, Ashland, OR, USA). Gating controls included Internal Negative Controls (INC) and Fluorescent Minus One (FMO) controls.

## Histological staining

For alkaline phosphatase staining, cells were fixed with 4% PFA for 30 min. Cells were washed twice with 1x PBS and then permeabilized with Tween 0.05% for 10 min. After permeabilization, cells were washed twice with 1x PBS, and then incubated with BCIP/NBT staining solution was added to the cells and incubated at room temperature for 10 min in the dark on a shaker. The cells were washed 3 times with deionized water before analysis. For Alizarin Red staining, fixed cells were incubated with Alizarin Red staining solution at room temperature in the dark for 45 min. The cells were washed 3 times with deionized water before analysis. For TRAP staining, Acid Phosphatase Leukocyte kit (Sigma Aldrich) was used following the manufacturer's instructions. Briefly, cells were fixed with 4% PFA for 30 min. 0.5 ml Fast Garnet GBC Base Solution and 0.5 ml Sodium Nitrite Solution were mixed by inversion for 30 s. After 2 min, the solution was mixed with 1.0 ml Naphthol AS-Bl Phosphate Solution, 0.5 ml Acetate Solution, and 2.0 ml Tartrate Solution. Cells were incubated for 1 h in 37 °C water bath protected from light. After 1 h, cells were rinsed thoroughly in deionized water, then counterstained 2 min in Hematoxylin Solution.

## Fluorescence staining

Primary cells grown on a glass-bottom well plate or the MPS were fixed by incubating with 4% paraformaldehyde in PBS for 30 min. Cells were washed three times with PBS for 5 min between every step. Cells were permeabilized with 0.2% Triton X-100 for 30 min at RT and blocked with 3% BSA + 0.1 M Glycine in PBS + 0.1% Tween overnight at 4 °C. Cells were incubated with 5 µg/ml anti-human CD31-CoraLite 594 (Proteintech, L594-11265), 5 mg/ml anti-TROP2-488 (R&D, AF650, goat), 5 mg/ml Anti-CD163-594 (Novus, BM4041AF594), 5 mg/ml anti-RANK-488 (Novus, 64C1385), 1 1:500 Texas Red®-X phalloidin, 1:1000 DAPI in 3% BSA overnight at 4 °C.

Adipocytes were labeled with LipidSpot (Biotium) following the manufacturer's instructions. Briefly, cells or devices were fixed with 4% PFA and incubated with 1X LipidSpot solution (1:1000 for 2D cells and 1:50 for devices) in PBS overnight at 4 °C.

Plates were imaged using a Nikon TI® Eclipse inverted microscope (Melville, New York), and devices were imaged in a Leica SP8 3X STED Super-resolution microscope (Wetzlar, Germany) in the UW-Madison Optical Imaging Core.

## RNA isolation and qPCR

RT-qPCR was used to analyze the expression of multiple genes in the Bone MPS. When prostate tumor spheroids were present in the side port, the hydrogel was cut, and the port was separated from the main chamber. Two devices were collected per condition, and RNA was isolating using RNA easy plus kit (Qiagen), and mRNA was reverse transcribed to cDNA using the RT2 PreAMP cDNA Synthesis Kit (330451, Qiagen). cDNA was analyzed by RT-qPCR using either Taqman or Qiagen RT2 profiler custom panel (CLAH25337, Qiagen). For RT2 profiler, data was analyzed using the Qiagen online software (http://pcrdataanalysis.sabiosciences.com/pcr/arrayanalysis.php). Only genes that exhibited statistically significant differences were discussed in the results section.

## Multiplex bead-based ELISA

Multiplexed protein secretion analysis was performed on media using the Bead-Based ELISA system MAGPIX (Luminex Corp.) with a custom panel from BD Biosciences. Briefly, conditioned media were retrieved from both control and metastatic bone marrow microphysiological systems at days 0, 3, and 7. Sample preparation and detection were performed according to the manufacturer's instructions. Data were collected with xPonent software (Luminex), and soluble factor concentrations in media were calculated using mean fluorescence intensities.

## Calcium quantification

Calcium was quantified using Stanbio Calcium (CPC) LiquiColor kit to provide a quantitative plate reader-based analysis. This was performed following manufacturer instructions. Briefly, 6 devices were pulled and digested overnight in 200 µL 1 M HCL at 60 °C and 10 rpm. Calcium standard solutions (0–800 ng/ml) were prepared from a stock calcium solution (100 µg/mL). A 96-well plate was used for the assay, with 10 µL of each standard or sample added to individual wells in triplicate, followed by 200 µL of working dye solution (prepared by mixing Reagent 1 and Reagent 2 in a 1:1 ratio). The plate was incubated for 10 min at room temperature, and absorbance was measured at 570 nm using a plate reader. Calcium content in the samples was determined using a standard curve.

## Second Harmonic Generation (SHG)

Collagen hydrogel structure was visualized by second harmonic (SHG) using a custom-built inverted multiphoton microscope (Bruker Fluorescence Microscopy, Middleton, WI), as described previously[88].

The system consisted of a titanium:sapphire laser (Spectra-Physics, Insight DS-Dual), an inverted microscope (Nikon, Eclipse Ti, Melville, New York, NY, USA), and a Nikon Apo 40×/1.25 WI λS objective. Collagen fibers were excited using an of 890 nm, an emission bandpass filter of 440/80 nm, and a GaAsP photomultiplier tube (H7422P-40, Hamamatsu). FAD fluorescence was then isolated using an excitation wavelength of 890 nm and an emission bandpass filter of 550/100 nm. 3 z-stacks with 20 slices (1 µm steps) were collected per device, starting at a 100 µm distance from the bottom of the gel, to avoid edge effects. Four devices per condition and biological replicate were imaged for posterior analysis. For the collagen architecture analysis, FAD intensity was subtracted from the SHG. Background subtraction (rolling ball in ImageJ, 50 pts) and despeckling algorithms were then applied to all stacks. From these stacks, we analyzed fiber shape descriptors (i.e., fiber width and length), fiber organization (i.e., fiber alignment and winding), and parameters associated with fiber density (i.e., void area, fibers per field of view, and gap or pore area). Fiber shape descriptors, fiber organization, and fibers per field of view were extracted by running 3 images per stack through CT-Fire and Curvealign software workflows (UW-Madison LOCI)[89–93]. Gap sizes from SHG images of the different collagen matrices were analyzed using Image J. Stacks were z-projected (median signal modality), and a threshold was applied to determine where no collagen fibers were present. The resulting 12 largest gaps per each projected image were outlined manually, and their area was measured.

## Statistics and reproducibility

Statistical analysis was performed using GraphPad Prism 10. A two-way ANOVA + Dunnett's multiple comparison test was performed to compare viability in co-culture conditions for Fig. 1. A one-way ANOVA + Tukey's post-hoc test was used in the experiments for Figs. 6, and 7, for comparisons across multiple time points or across treatment conditions. Two-tailed

unpaired *t*-tests with Welch's or Mann-Whitney correction were used in experiments for Figs. 1, 3, 4–6, 8 and Supplementary Fig. 2 when comparing changes in gene expression between co-culture environments and for characterization of ECM modifications noted on SHG. All error bars in figures indicate the mean ± s.d. across normalized technical replicates across at least $n = 3$ biological replicates using 95% confidence interval.

## Reporting summary
Further information on research design is available in the Nature Portfolio Reporting Summary linked to this article.

## Data availability
The experimental data results that support the findings of this study are available in EveAnalytics database under the study names "Sacituzumab-Govitecan killing in the bone metastasis chip with patient PCa" (https://eve.eveanalytics.com/assays/assaystudy/1464/), "Effect of the stroma on Trop-2 ADC in primary samples" (https://eve.eveanalytics.com/assays/assaystudy/1466/), "Effect of Tro2-ADC on the bone marrow stroma" (https://eve.eveanalytics.com/assays/assaystudy/1467/), "Trop2 ADC PCa Cell lines in Bone Marrow MPS" (https://eve.eveanalytics.com/assays/assaystudy/1463/), "Cell phenotype in multiphenotype media" (https://eve.eveanalytics.com/assays/assaystudy/1458/), "Phenotype characterization of stroma cells in the bone MPS" (https://eve.eveanalytics.com/assays/assaystudy/1459/), "Protein analysis of bone marrow MPS in presence and absence of Prostate cancer epithelial cells" (https://eve.eveanalytics.com/assays/assaystudy/1460/), "Conventional treatments for prostate cancer on the Bone Marrow MPS" (https://eve.eveanalytics.com/assays/assaystudy/1461/), "The effect of the stroma in effectivity of conventional treatments against prostate cancer" (https://eve.eveanalytics.com/assays/assaystudy/1462/), "Mineralization over time" (https://eve.eveanalytics.com/assays/assaystudy/1448/), "qPCR bone Chips controls vs tumor" (https://eve.eveanalytics.com/assays/assaystudy/1418/) "Collagen remodeling" (https://eve.eveanalytics.com/assays/assaystudy/1457/), "ADC diffusion assay" (https://eve.eveanalytics.com/assays/assaystudy/878/), "Bone marrow microenvironment viability patient 18" (https://eve.eveanalytics.com/assays/assaystudy/675/), "Bone marrow microenvironment viability patient 22" (https://eve.eveanalytics.com/assays/assaystudy/748/), "Media optimization for Prostate Cancer device" (https://eve.eveanalytics.com/assays/assaystudy/643/). Single cells data are not publicly available to maintain the protection of patient privacy. Data sharing requests must be submitted to the University of Wisconsin-Madison for review and approval.

## Code availability
The code used for single-cell RNA analysis is available on GibHub at the link https://github.com/fauzan-ahmed/CToC.git.

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

## Acknowledgements

This work was supported by the University of Wisconsin, Carbone Cancer Center Support Grant NIH P30CA014520, Medical Scientist Training Program Grant T32 GM140935, The Veterans Affairs Advanced Fellowship in Women's Health, Kirschstein National Research Service Award T32 HL07899, 2019 PCF Challenge Award 19CHAL12, Prostate SPORE P50CA269011, NIH UG3CA260692, NIH R01CA206458. The BD LSR II Fortessa instrument was supported by the NIH Shared Instrumentation Grant Special BD LSR Fortessa #1S100D018202-01. The authors thank the University of Wisconsin Carbone Cancer Center Translational Science Biocore BioBank, Circulating Biomarker Core, and Flow Cytometry Core, supported by P30 CA014520, for use of their facilities and services. We would also like to thank the University of Wisconsin Optical Imaging Core for their help in image collection. We thank Onexio Biosystems for their microDUO culture platform. The authors thank Dr. David Jarrard for collecting the primary prostate tissue and Dr. Wei Huang for the pathological evaluation of the tissue. We thank Gilead Sciences Inc. for supplying sacituzumab govitecan (no funding was provided for the study).

## Author contributions

S.C.K., D.J.B. and J.M.L. conceived the research. C.S.D. and R.C.Y. designed the experiments. C.S.D., R.C.Y., N.S., M.V.M., P.G.G., A.B.D., F.A. and E.H. carried out the experiments and analyzed the data. A.B.D. and N.L. prepared and characterized the cells. E.B. and K.S. fabricated LumeNEXT devices. S.M.P., M.N.S. provided the methodology and assisted with SHG analysis. C.S.D. and R.C.Y. prepared the manuscript and all authors provided feedback. S.C.K., D.J.B. and J.M.L. provided supervision and contributed to edits of the manuscript.

## Competing interests

The authors declare the following competing interests: D.J.B. holds equity in Bellbrook Labs LLC, Tasso Inc., Salus Discovery LLC, Lynx Biosciences Inc., Stacks to the Future LLC, Flambeau Diagnostics LLC and Onexio Biosystems LLC. D.J.B. is also a consultant for Abbott Laboratories. J.M.L. served as paid consultant/received honoraria from Sanofi, AstraZeneca, Gilead, Pfizer, Astellas, Seattle Genetics, Janssen and Immunomedics. N.S. served as paid advisory board consultant to Amgen. M.N.S. reports institutional research support from Novartis unrelated to the current study.

## Ethical approval

Inclusion and Ethics: All patients provided written, informed consent under an Institutional Review Board (IRB) approved protocol at the University of Wisconsin-Madison. All ethical regulations relevant to human research participants were followed.
