## [Transparent Peer Review file · Communications Biology]

Engineering the bone metastatic prostate cancer niche through a microphysiological system to report patient-specific treatment response

Corresponding Author: Dr Sheena Kerr

This manuscript has been previously submitted to another journal. This document only contains information relating to versions considered at Communications Biology.

Version 0:

Reviewer comments:

Reviewer #1

(Remarks to the Author)

The authors developed a multicellular microphysiological device composed of 8 different cell types, mostly patient-derived, which were shown to survive up to 7 days in co-culture and maintain their phenotype. This platform was then applied to perform proof-of-principle studies on the response to chemotherapy and targeted therapy, showing increased resistance in the presence of bone stromal cells. The concept of MPS is novel and the ability to combine so many different cell populations is exciting. However, the following concerns should be addressed.

1. Line 271: Viability on day 4 and 7 was defined as long-term cell culture, but 1 week of culture is not really long-term culture.
2. Single-cell RNA sequencing analysis results are vague and should be investigated further. The relevance of the cluster marked as unknown is unclear. Further analysis of specific transcripts should help to better define this cluster.
3. There is no evidence of cell proliferation (line 273), but an increase in the number of osteoblasts; how do the authors explain this?
4. There is a discrepancy between the amount of osteoclasts, 1% (line 309; Figure 2A) vs 10% (line 313). How do the authors justify this? Osteoclasts should not proliferate; furthermore, the cells shown in Figure 3E are not multinucleated as shown in Figure S1D. Do osteoclasts change morphology in MPS?
5. Images showing progressive increase in mineralization should be reported in Figure 3F.
6. Images at day 0 and 7 in Supplementary Figure 2A look very similar. Differences and parameters analyzed should be highlighted more clearly. Description of how the analysis was performed should be included in the methods.
7. The significance of bulk RNA sequencing analysis is unclear, as gene expression analysis can be retrieved from single-cell RNA sequencing analysis.
8. There is no reference to Figures 4C and D in the text.
9. The authors should discuss more extensively other state-of-the-art bone mimics/in vitro engineered cultures and how MPS compares, including advantages and disadvantages. They should also discuss further feasibility for personalized medicine, stability of the system at longer time intervals, comparison of stromal cells derived from different patients and their impact on response to therapy.

Reviewer #2

(Remarks to the Author)

This study is commendably well-planned and executed, showcasing a strong foundation in its methodology and approach. However, the manuscript falls short in terms of scientific rigor, particularly in the introduction and discussion sections. A significant concern is that many of the cited references do not directly relate to the core hypotheses being explored in this research.

To enhance the manuscript's relevance and depth, the authors would benefit from examining landmark publications by two of the leading research groups specializing in the engineering of microenvironments for prostate cancer and its metastasis. Specifically, the authors should consider important works from the Risbridger group at Monash University and the Clemens group at Queensland University of Technology. These groups have established a strong reputation in this field, and their insights could provide a crucial context for the current study.

By thoroughly engaging with these foundational publications, the authors can refine their introduction and discussion, ensuring that they align more closely with recognized research in the area. This endeavor will not only strengthen their manuscript but will also contribute to a more articulate and focused presentation of their findings and their significance within the broader scientific discourse.

Reviewer #3

(Remarks to the Author)

Sánchez-de-Diego et al. suggested an innovative approach by co-culturing various types of stromal cells to construct an in vitro bone metastatic prostate cancer niche MPS model and evaluate drugs. By using patient-derived primary cells and testing different types of drugs, they demonstrated the potential of this platform as a patient-specific evaluation system for personalized medicine. However, there are major concerns regarding the manuscript as listed below.

Major comments:

1. As mentioned in the introduction, additional experiments might be necessary to show how prostate cancer affects osteoblasts and osteoclasts. A biological assay examining the interaction between prostate cancer and bone stromal cells would help address this. Otherwise, it's hard to say this MPS really represent the prostate cancer bone metastatic microenvironment.
2. In Figure 3C, it shows an increase in osteoblasts and M2, along with a slight decrease in endothelial cells on Day 7. The reason behind these changes isn't clear, and it's unclear why no replication experiments were done to confirm these observations.
3. In addition to Figure 3F, if there were quantitative data on calcium deposition, such as staining images, the data would be much clearer.
4. The discussion describes this MPS as a platform for long-term culture, but in Fig. 3 and other experiments, the co-culture period using the chip was limited to 7 days. Is there any reason for the culture limitation?
5. Regarding the data interpretation, the PPAR γ results (Line 332) don't seem to show an increasing trend as mentioned in the manuscript. Similarly, the PSA expression levels in LNCaP cells (Lines 355 and 356) don't appear higher than other PCa cell lines. For accurate comparisons, negative controls should be provided along with a quantitative analysis of the data.
6. In the Discussion section and Figures 5 and 7, the authors showed that reduced chemotherapeutic efficacy is due to the bone chip environment but didn't explain the relevant mechanisms. Additional experiments or in-depth discussion of the results are needed to address this (e.g., changes in hydrogel physical properties, drug delivery efficiency, or cancer cell-stromal cell interactions).
7. In Fig. 6, the drug concentrations were optimized in a 2D cell environment, but the actual target was the spheroid encapsulated within the hydrogel. Shouldn't the drug concentration optimization have been done with the spheroids at least? Additionally, it would be better to optimize the spheroid fabrication process to ensure consistent sizes for evaluating multiple drugs.
8. In Figure 7D, it would be helpful to include the data for SG treatment at 5 $\mu\text{g/ml}$ for comparison. For example, Figure 7B shows that Donor 3 had similar effects at 5 $\mu\text{g/ml}$ and 10 $\mu\text{g/ml}$. It's important to determine whether this pattern changes under bone microenvironment conditions.

Minor comments:

1. This manuscript has tons of errors, mistakes and grammatical errors. For example, Line 323 refers to Supplementary Figure 2B, but it should actually be Supplementary Figure 2A, and Line 768 uses "oC" when it should be "C." Unless all these issues are thoroughly addressed, this manuscript cannot be published.
2. For Figure 1D, the statistical analysis of the adipocyte data should be double-checked to ensure accuracy.
3. In the legend in Supplementary Figure 1A "osteoclast" should be corrected to "osteoblast." Also, it seems that the alizarin red solution wasn't fully washed off, so a completely washed and clear image is needed to accurately interpret the data.

Version 1:

Reviewer comments:

Reviewer #1

(Remarks to the Author)

I think the authors have responded satisfactorily to my requests.

Reviewer #2

(Remarks to the Author)

The authors addressed all reviewers' comments satisfactorily, and the work can be accepted.

Reviewer #3

(Remarks to the Author)

I recommend that it be published, as all concerns have been satisfactorily addressed.

Reviewers' comments: We thank the Reviewers for their thorough and insightful critique that provided an opportunity to improve the quality of our manuscript. Please find below a point-by-point response to the Reviewers' comments. We have revised the manuscript to incorporate the Reviewers' suggestions and highlighted all changes in the manuscript.

Reviewer 1:

The authors developed a multicellular microphysiological device composed of 8 different cell types, mostly patient-derived, which were shown to survive up to 7 days in co-culture and maintain their phenotype. This platform was then applied to perform proof-of-principle studies on the response to chemotherapy and targeted therapy, showing increased resistance in the presence of bone stromal cells. The concept of MPS is novel and the ability to combine so many different cell populations is exciting. However, the following concerns should be addressed.

Critique 1: *Line 271: Viability on day 4 and 7 was defined as long-term cell culture, but 1 week of culture is not really long-term culture.*

Response 1: We appreciate the Reviewer's feedback and have removed the reference to "long-term cell culture" in our manuscript, referencing 7 days as sufficient for measuring treatment response. We have also added text to the discussion to suggest extending the culture time as a good direction for further studies.

Critique 2: *Single-cell RNA sequencing analysis results are vague and should be investigated further. The relevance of the cluster marked as unknown is unclear. Further analysis of specific transcripts should help to better define this cluster.*

Response 2: We agree with the Reviewer's suggestion and have further refined our single-cell RNA sequencing analysis. We performed additional analysis of the unknown population as a subcluster. The cells within this subcluster demonstrated a lower level of gene expression than the cells in the other analyzed clusters. Gene expression was largely restricted to housekeeping genes, suggesting this population may be mainly damaged cells. To further identify the cells in the cluster, we identified highly variable genes and compared these to marker genes from bone stromal lineages. This analysis suggests these cells are likely MSC that are early in the differentiation process and beginning to commit to lineages. We have modified Figure 3 and the text accordingly with these additional analyses. Additionally, we have included a new supplementary figure detailing the mixed population within this cluster. To enhance clarity, we have also specified the transcripts used to define each cluster in the Results Section (Page 10).

Critique 3: *There is no evidence of cell proliferation (line 273), but an increase in the number of osteoblasts; how do the authors explain this?*

Response 3: The Reviewer raises a valuable question. Our data indicate that the observed increase in osteoblast numbers is likely due to the ongoing differentiation of MSCs rather than cell proliferation. To improve interpretation, we have modified the Results (Page 10) and Discussion (Page 25-26) to indicate that these observations are also supported by the reduction in MSC-related gene expression captured in our qPCR analysis. Furthermore, our single-cell RNA sequencing analysis identified a population of MSC in a transitional state towards osteoblast and adipocytes.

Critique 4: *There is a discrepancy between the amount of osteoclasts, 1% (line 309; Figure 2A) vs 10% (line 313). How do the authors justify this? Osteoclasts should not proliferate; furthermore, the cells shown in Figure 3E are not multinucleated as shown in Figure S1D. Do osteoclasts change morphology in MPS?*

Response 4: We thank the Reviewer for this valuable comment and agree that this point was not conveyed clearly in the original manuscript and needed further clarification. While early in differentiation osteoclasts may remain mononuclear, we agree that fully differentiated osteoclasts should be multinucleated. We have included improved high magnification images in Figure 3E showing that these cells are multinucleated in the MPS. While we aimed to incorporate 1% osteoclasts in our MPS, the starting population is not 100% pure as they are

differentiated from monocytes, resulting in a mixed population of M2 macrophages and osteoclasts, and effective sorting techniques were not available to isolate pure differentiated osteoclasts. Further, patient heterogeneity also contributes to differences in differentiation rates and final osteoclast numbers. Therefore, while we try to standardize adding 1% osteoclasts to the MPS to reflect physiological composition there may be some variability when adding this small number of cells to the chip. We have included this information in Results (Page 10) and Discussion (Page 25-26) sections of the revised manuscript and edited the text to clarify that RANK⁺ cells comprised approximately 2% to 9% of the total cell population within the MPS” (Page 10).

***Critique 5:** Images showing progressive increase in mineralization should be reported in Figure 3F.*

Response 5: We appreciate the Reviewer’s suggestion and agree for the need to improve clarity in our description of the mineralization methodology. To quantify calcium, the gels were dissolved and analyzed using the StanBio Calcium Liquicolor test, which is then measured on a plate reader, rather than an image-based method. We have added this information to the Results section to specify (Page 10). Additionally, we have further clarified this approach in the Methods section (Page 31).

***Critique 6:** Images at day 0 and 7 in Supplementary Figure 2A look very similar. Differences and parameters analyzed should be highlighted more clearly. Description of how the analysis was performed should be included in the methods.*

Response 6: We appreciate the Reviewer’s suggestions and have improved the description in the Methods section for clarity as to how the analysis was performed for these assays. Additionally, we have carefully revised the Methods section to ensure all relevant details for the methodology are included. The collagen fibers are too small to appreciate changes by conventional observational methods. Therefore, for image analysis, we used previously established image analysis programs to measure fiber parameters including CurveAlign (Schneider et al., Bredfeldt et al., 2014a; Liu et al., 2017; Liu et al, 2020), and CT-Fire (2020 Bredfeldt et al., 2014b). These software workflows were developed at University of Wisconsin-Madison Laboratory for Optical and Computational Instrumentation <https://loci.wisc.edu/curvealign/>. Additionally, gap sizes from Second-Harmonic Generation (SHG) images of the different collagen matrices were analyzed using ImageJ software and this methodology has been updated.

Schneider, C.A., Pehlke, C.A., Tilbury, K., Sullivan, R., Eliceiri, K.W., and Keely, P.J. (2013). Quantitative Approaches for Studying the Role of Collagen in Breast Cancer Invasion and Progression. In Second Harmonic Generation Imaging, F.S. Pavone, and P.J. Campagnola, eds. (New York: CRC Press), p. 373.

Bredfeldt, J.S., Liu, Y., Pehlke, C.A., Conklin, M.W., Szulczewski, J.M., Inman, D.R., Keely, P.J., Nowak, R.D., Mackie, T.R., and Eliceiri, K.W. (2014a). Computational segmentation of collagen fibers from second-harmonic generation images of breast cancer. *J. Biomed. Opt.* 19, 016007–016007. PMID: 24407500

Bredfeldt, J.S., Liu, Y., Conklin, M.W., Keely, P.J., Mackie, T.R., and Eliceiri, K.W. (2014b). Automated quantification of aligned collagen for human breast carcinoma prognosis. *J. Pathol. Inform.* 5. PMID: 25250186

Liu, Y., Keikhosravi, A., Mehta, G.S., Drifka, C.R., and Eliceiri, K.W. (2017). Methods for quantifying fibrillar collagen alignment. In *Fibrosis: Methods and Protocols*, L. Rittié, ed. (New York: Springer). PMID: 28836218

Liu, Y., Keikhosravi, A., Pehlke, C. A., Bredfeldt, J. S., Dutson, M., Liu, H., Mehta, G. S., Claus, R., Patel, A. J., Conklin, M. W., Inman, D. R., Provenzano, P. P., Sifakis, E., Patel, J. M. & Eliceiri, K. W. (2020). Fibrillar collagen quantification with curvelet transform based computational methods. *Front. Bioeng. Biotechnol.* 8, 198. PMID: 32373594

***Critique 7:** The significance of bulk RNA sequencing analysis is unclear, as gene expression analysis can be retrieved from single-cell RNA sequencing analysis.*

Response 7: We thank the Reviewer for this valuable comment and for the opportunity to clarify the use of the term "bulk RNA from the devices" in our manuscript. We originally intended to convey that we obtained RNA from pooled devices and performed RT qPCR analysis. To improve clarity, have removed the word "bulk" and specified that this was a RT-qPCR analysis.

While scRNA sequencing provides valuable insights into gene expression at the single-cell level, RT-qPCR was used to confirm the results because it allows for the quantification of specific transcripts in a more targeted

and reproducible manner. Furthermore, this approach provides validation of our single-cell data and offers a higher degree of sensitivity for detecting gene expression changes with a well-defined set of cell type specific biomarkers.

Critique 8: There is no reference to Figures 4C and D in the text.

Response 8: We thank the Reviewer for pointing to this oversight. We have replaced references to Figures 4C and D in the text to ensure proper alignment between the results and the figures.

Critique 9: The authors should discuss more extensively other state-of-the-art bone mimics/in vitro engineered cultures and how MPS compares, including advantages and disadvantages. They should also discuss further feasibility for personalized medicine, stability of the system at longer time intervals, comparison of stromal cells derived from different patients and their impact on response to therapy.

Response 9: We appreciate the Reviewer's suggestion to expand on the comparison between our microphysiological system (MPS) and other state-of-the-art bone mimics or engineered cultures. We have addressed this comparison in the Introduction (Page 3) and Discussion (Page 24) of the revised manuscript, highlighting both the advantages and limitations of our MPS system relative to other existing models of the bone microenvironment.

Specifically, we have expanded discussion of other state of the art bone models including humanized mouse models, such as the humanized tissue-engineered bone construct (hTEBC) in NSG mice, and discussed advantages and disadvantages, which include species-specific limitations, differences in immune system compatibility and the influence of murine physiological factors. We have also provided more context for our MPS with other bioengineered human systems. Most importantly, our fully human system includes multiple bone stromal cell types allowing study of the multicellular communication in that niche and allows real-time monitoring of dynamic processes such as tumor-stroma interactions, osteoclastogenesis, and therapy-induced changes in the bone microenvironment in a model system that is translationally relevant to human physiology (Page 3).

Additionally, we have expanded the Discussion (Page 24, 26) regarding feasibility and advantages of the MPS for personalized medicine, and future plans to incorporate longer term cultures and matched patient-derived tumor cells and stromal components. However, we do acknowledge that patient-derived samples can introduce variability. We have used and compared stromal cells from different donors in co-culture with the same tumor cells and did not observe significant changes in response to the treatments tested. However, using fully matched patient models or different treatments could yield different results.

Reviewer 2:

Critique 1: This study is commendably well-planned and executed, showcasing a strong foundation in its methodology and approach. However, the manuscript falls short in terms of scientific rigor, particularly in the introduction and discussion sections. A significant concern is that many of the cited references do not directly relate to the core hypotheses being explored in this research.

To enhance the manuscript's relevance and depth, the authors would benefit from examining landmark publications by two of the leading research groups specializing in the engineering of microenvironments for prostate cancer and its metastasis. Specifically, the authors should consider important works from the Risbridger group at Monash University and the Clemens group at Queensland University of Technology. These groups have established a strong reputation in this field, and their insights could provide a crucial context for the current study.

By thoroughly engaging with these foundational publications, the authors can refine their introduction and discussion, ensuring that they align more closely with recognized research in the area. This endeavor will not only strengthen their manuscript but will also contribute to a more articulate and focused presentation of their findings and their significance within the broader scientific discourse.

Response 1: We thank the Reviewer for the in-depth review of our manuscript and positive comments. We appreciate the thoughtful suggestions to improve interpretation of our work and give us the opportunity to increase the overall quality of our manuscript through their recommendation to incorporate landmark publications from the Risbridger group at Monash University and the Clemens group at Queensland University of Technology. In response, we have reviewed and integrated key insights from these influential research groups into both the Introduction (Page 3) and Discussion sections (Pages 24) of the manuscript. This has provided a more comprehensive context for our study.

Reviewer 3:

Sánchez-de-Diego et al. suggested an innovative approach by co-culturing various types of stromal cells to construct an in vitro bone metastatic prostate cancer niche MPS model and evaluate drugs. By using patient-derived primary cells and testing different types of drugs, they demonstrated the potential of this platform as a patient-specific evaluation system for personalized medicine. However, there are major concerns regarding the manuscript as listed below.

Major comments:

Critique 1: *As mentioned in the introduction, additional experiments might be necessary to show how prostate cancer affects osteoblasts and osteoclasts. A biological assay examining the interaction between prostate cancer and bone stromal cells would help address this. Otherwise, it's hard to say this MPS really represent the prostate cancer bone metastatic microenvironment.*

Response 1: We thank the Reviewer for the in-depth review and valuable feedback to improve the quality of our manuscript. We have conducted additional experiments to interrogate the effects of prostate cancer (PCa) cells on the bone stromal cells within our MPS. Specifically, we utilized RT-qPCR and MagPix to analyze gene and protein expression changes following the addition of primary prostate cancer tumor spheroids into the system. We have incorporated a new figure 5 to illustrate these findings.

Our qPCR analysis using the RT2 Osteogenesis Profiler array, analyzed 84 genes. We found significant changes in genes associated with osteoblast and osteoclast differentiation. Notably, the presence of primary prostate tumor cell spheroids led to upregulation of ALPL, a marker for osteoblast differentiation, as well as GLI1 and TWIST1, which are linked to cell differentiation, proliferation, and survival. These are consistent the osteoblastic phenotype observed in prostate cancer. Additionally, we observed upregulation of CTSK, an osteoclast marker involved in bone resorption and tumor invasiveness. Interestingly, we also saw downregulation of BMP4, ACVR1, and TGFB1, which are important components of the bone morphogenic protein signaling pathway implicated in cancer progression.

We further assessed protein expression levels using a multiplexed bead-based ELISA and observed distinct changes in protein levels over time. By day 7 of co-culture, we saw significant alterations in various growth factors and cytokines. Specifically, the levels of TGF- α , BMP-9, and VEGF-A increased, while DKK1 and OPG were downregulated, highlighting the impact of primary prostate tumor cell spheroids on stromal cell activity and the bone microenvironment. Additionally, we found changes in the levels of M-CSF, GM-CSF, and MMP-2, further confirming the reciprocal signaling occurring between the prostate cancer cells and the bone stromal cells.

These findings underscore the complex interactions between prostate cancer cells and the bone microenvironment, and we believe they are a good addition to support the validity of our MPS as a representation of the prostate cancer bone metastatic niche.

Critique 2. *In Figure 3C, it shows an increase in osteoblasts and M2, along with a slight decrease in endothelial cells on Day 7. The reason behind these changes isn't clear, and it's unclear why no replication experiments were done to confirm these observations.*

Response 2: We thank the Reviewer for pointing this out and understand the Reviewer's concern. To further confirm and validate our observations, we added qPCR analysis to assess gene expression changes, which helped corroborate our findings from the single cell sequencing. With respect to the increase in osteoblasts and M2 macrophages, we see evidence for continued differentiation of cells within the microphysiological system, which would contribute to the observed changes. The qPCR data provided additional support for these

findings. As the data are shown as a percentage of the total, the minor decrease in the percentage of endothelial cells is likely a result of increased percentages of differentiated cells.

Critique 3: In addition to Figure 3F, if there were quantitative data on calcium deposition, such as staining images, the data would be much clearer.

Response 3: We thank the Reviewer for this valuable suggestion. We appreciate your feedback on the need for quantitative data on calcium deposition. To clarify, we used the Stanbio Calcium (CPC) LiquiColor kit to assess calcium deposition, which is a colorimetric method established to measure the concentration of calcium in biological samples, such as culture media or tissue homogenates. This assay utilizes a specific reagent that reacts with calcium ions, resulting in a color change. The intensity of the color change is directly proportional to the calcium concentration, which can be quantified using a spectrophotometer. The gels need to be digested for this analysis, therefore, images are not available. However, we believe this assay platform provides quantification of mineralization in our system, and we have integrated further clarification of this methodology in the Results section (Page 10) to improve interpretation.

Critique 4: The discussion describes this MPS as a platform for long-term culture, but in Fig. 3 and other experiments, the co-culture period using the chip was limited to 7 days. Is there any reason for the culture limitation?

Response 4: We appreciate the Reviewer's feedback and have removed the reference to "long-term cell culture" in our manuscript. 7 days was used as this provided sufficient culture time to measure treatment responses. We have added additional text to discuss potential for longer term culture as a future direction for the platform.

Critique 5: Regarding the data interpretation, the PPAR γ results (Line 332) don't seem to show an increasing trend as mentioned in the manuscript. Similarly, the PSA expression levels in LNCaP cells (Lines 355 and 356) don't appear higher than other PCa cell lines. For accurate comparisons, negative controls should be provided along with a quantitative analysis of the data.

Critique 5: We thank the Reviewer for pointing out the error in the Results section, which has been revised and corrected. Additionally, we have modified Figure 4A to include negative controls and quantitative median fluorescence intensity (MFI) read-outs for the flow cytometry data to provide more accurate comparisons and ensure the reliability of our findings. This addition enhances the clarity and robustness of the data analysis. Furthermore, upon revising the flow data, we identified an error in the representative histogram shown for expression levels of TROP-2 for donor 1. We have corrected this error in the Figure 4A and modified the Results (Page 13), accordingly.

Critique 6: In the Discussion section and Figures 5 and 7, the authors showed that reduced chemotherapeutic efficacy is due to the bone chip environment but didn't explain the relevant mechanisms. Additional experiments or in-depth discussion of the results are needed to address this (e.g., changes in hydrogel physical properties, drug delivery efficiency, or cancer cell-stromal cell interactions).

Response 6: We thank the Reviewer for the valuable suggestion. This is an area of ongoing work in the laboratory that is beyond the scope of our current study, but we anticipate discussing in detail in a future manuscript. We agree with the Reviewer that this is an important point so have expanded our discussion (Page 25) on the mechanisms underlying reduced chemotherapeutic efficacy in the bone microenvironment adding the following text:

"Further studies will investigate in depth the mechanisms underlying chemotherapy resistance in the bone microenvironment. One of the primary mechanisms that might be responsible for reducing chemotherapy efficacy in the bone is alterations in ECM, including changes in collagen deposition and increased bone mineralization, which creates a physical and biochemical barrier that limits drug penetration^{52,75,76}. Additionally, cancer cells within the bone microenvironment can exploit signaling pathways that enhance survival and promote drug resistance. We observed that the presence of tumor cells upregulated the expression of TGF- α and EGF, both of which are abundantly released during osteoclastic bone resorption (Figure 5)⁷⁷. TGF- α plays a crucial role in stimulating tumor growth and enhancing resistance to therapy by activating the EGFR pathway⁷⁸. This leads to downstream activation of survival signaling cascades, including MAPK/ERK and PI3K/Akt, which promote tumor proliferation and inhibit apoptosis^{78,79}. Furthermore, TGF- α and EGF signaling drive epithelial-to-mesenchymal transition (EMT), a process that enhances tumor cell motility, invasion, and

resistance to chemotherapy. In MPS containing prostate tumor spheroids we observed upregulation of EMT-related transcription factors GLI and TWIST1, compared to MPS without tumors. These results further reinforce a mesenchymal, drug-resistant phenotype. GLI, a key effector of Hedgehog signaling, has been implicated in maintaining cancer stem-like properties, while TWIST1 plays a crucial role in EMT induction and resistance to apoptosis (Figure 5)^{80,81}. Activation of these pathways confers resistance to chemotherapy by promoting stemness, increasing DNA repair capacity, and suppressing pro-apoptotic signaling⁸².”

***Critique 7:** In Fig. 6, the drug concentrations were optimized in a 2D cell environment, but the actual target was the spheroid encapsulated within the hydrogel. Shouldn't the drug concentration optimization have been done with the spheroids at least? Additionally, it would be better to optimize the spheroid fabrication process to ensure consistent sizes for evaluating multiple drugs.*

Response 7: We thank the Reviewer for these insightful comments. The drug concentrations used in Figures 6-7 were based on clinical and *in vivo* data, and the dose response was tested in 2D cell culture to permit higher throughput and a larger number of conditions to be tested simultaneously. After narrowing down the concentrations, we did further optimization in 3D spheroid cell lines to confirm that these doses worked in 3D (Supplementary Figure 4). These drug concentrations were chosen to reflect realistic therapeutic levels, ensuring that our findings are translatable to clinical applications.

Regarding the spheroid fabrication process, we utilize a self-assembly method and the size of the spheroid can vary between cell lines based on the cell size. We believe this inherent variability in size is a strength as it mimics the tumor foci heterogeneity observed *in vivo*. To ensure robust results, we performed multiple experiments with spheroids of different sizes, and we did not observe significant differences in drug response based on spheroid size. The results represent an average response after evaluation of a representative number of spheroids, which helps account for this variability. Therefore, we believe that the approach we used to evaluate drug efficacy is relevant to understanding how these drugs perform in a more heterogeneous, *in vivo*-like environment. We have added text in the Discussion (Page 24) to illustrate this.

***Critique 8:** In Figure 7D, it would be helpful to include the data for SG treatment at 5 µg/ml for comparison. For example, Figure 7B shows that Donor 3 had similar effects at 5 µg/ml and 10 µg/ml. It's important to determine whether this pattern changes under bone microenvironment conditions.*

Response 8: We appreciate the Reviewer's valuable feedback. For the studies in Figure 7D we used 10 µg/ml SG since that dose elicited a response in PCa cells from all the donors tested in the MPS with a focus on comparing the presence versus absence of stromal cells in the bone microenvironment. While we agree with the Reviewer that including data for SG treatment at 5 µg/ml would certainly provide additional insights, we were unable to perform additional dose-response experiments on donor 3 as we did not have enough cells from this clinical biospecimen to perform these analyses. However, the results from Figure 7B, showing similar effects at 5 µg/ml and 10 µg/ml in Donor 3, suggest that the stromal influence may be the primary factor in determining therapeutic outcomes under these conditions.

Minor comments:

***Critique 1:** This manuscript has tons of errors, mistakes and grammatical errors. For example, Line 323 refers to Supplementary Figure 2B, but it should actually be Supplementary Figure 2A, and Line 768 uses “oC” when it should be “°C.” Unless all these issues are thoroughly addressed, this manuscript cannot be published.*

Response 1: We appreciate the Reviewer pointing out these errors including the reference to Supplementary Figure 2B and the use of “oC” instead of “°C.” We have thoroughly reviewed the manuscript and corrected these issues, along with addressing other minor grammatical and formatting mistakes.

***Critique 2:** For Figure 1D, the statistical analysis of the adipocyte data should be double-checked to ensure accuracy.*

Response 2: We thank the Reviewer for the comment. We have re-checked the statistical analysis for the adipocyte data in Figure 1D. Upon review, we confirm that the p-value is 0.12112, which indicates no significant difference.

Critique 3: *In the legend in Supplementary Figure 1A "osteoclast" should be corrected to "osteoblast." Also, it seems that the alizarin red solution wasn't fully washed off, so a completely washed and clear image is needed to accurately interpret the data.*

Response 3: We thank the Reviewer for pointing to this oversight. We have corrected the legend in Supplementary Figure 1A to refer to "osteoblast" instead of "osteoclast." Additionally, as suggested by the Reviewer, we have replaced the image with a better quality image for more accurate interpretation of the data.